# NeuroLangSeg: Language-Guided Subcortical Segmentation with Pseudo-Supervision and Anatomical–Linguistic Validation

**Ruiying Liu**[*1] [iD]                  RLIU60@EMORY.EDU
[1] *Department of Biomedical Informatics, Emory University*

**Jialu Liu**[*1]                  LIUJIALU2001@GMAIL.COM
**Xuzhe Zhang**[2]                  XUZHE.Z@COLUMBIA.EDU
[2] *Department of Biomedical Engineering, Columbia University*

**Chuang Huang**[3]                  CHUAN.HUANG@EMORY.EDU
[3] *Department of Radiology and Imaging Sciences, Emory University*

**Yun Wang**[1,4]                  YUN.WANG2@EMORY.EDU
[4] *Department of Computer Science, Emory University*

**Editors:** Accepted for publication at MIDL 2026

## Abstract

Recent advances in vision–language models and LLMs have introduced contextual anatomical reasoning into brain MRI segmentation. However, the field still suffers from a fundamental limitation: the absence of a unified anatomical definition of the structures being segmented. Existing datasets rely on labels produced by heterogeneous manual workflows, often lacking explicit anatomical criteria or consistent annotation standards. As a result, models learn and evaluate within isolated labeling systems, limiting cross-model comparison and valid anatomical measurements. To address these challenges, we introduce **NeuroLangSeg**, a language-guided framework that enforces a consistent anatomical protocol for subcortical segmentation. A key component of the framework is an anatomical–linguistic evaluator that acts as a training discriminator, encouraging the model to produce outputs by assessing shape characteristics, protocol-defined spatial relationships, and age- and sex-adjusted volumetric norms. Building upon this constraint, NeuroLangSeg integrates a pretrained image encoder with protocol-aligned anatomical prompts and a masked pseudo-labeling strategy, enabling data-efficient and interpretable learning under limited supervision. Together, these components yield anatomically consistent segmentations and support subject-level reporting grounded in a unified anatomical standard. Evaluation across diverse MRI datasets—including comparisons with state-of-the-art models—shows that NeuroLangSeg achieves +4.1 DSC / +8.0 NSD in in-site settings and +3.6 DSC / +14.5 NSD in cross-site generalization over the average baseline, enabled by its LLM–visual integration, while delivering anatomically verifiable predictions suitable for both research and clinical use. GitHub: https://github.com/jlliu2001/SAT_MPL

**Keywords:** Anatomical Protocol, Language-Driven Segmentation, Anatomical–Linguistic Evaluation, Brain MRI

---

[*] Contributed equally

## 1. Introduction

Accurate segmentation of subcortical brain structures is fundamental to quantitative analysis and clinical assessment. Most regions such as the hippocampus, amygdala, and thalamus, enable detailed investigations of brain development, aging, and neuropathology, supporting downstream analyses of structure–function relationships and population-level biomarkers (Baribeau et al., 2019; Cruz et al., 2023; de Jong et al., 2008). Although manual delineation remains the gold standard for defining anatomical boundaries, it is time-consuming, labor-intensive, and dependent on expert knowledge.

While traditional neuroimaging pipelines such as FreeSurfer (Fischl, 2012), BrainSuite (Kim et al., 2024), ANTs (Avants et al., 2011), and FSL (Jenkinson et al., 2012) have been widely used for automated frameworks for structural analysis, their multi-stage registration and optimization procedures are computationally intensive and difficult to scale for large datasets or clinical workflows. In contrast, recent advances in deep learning have substantially improved medical image segmentation, allowing models to learn rich representations directly from MRI data and achieve high accuracy across diverse anatomical and clinical tasks (Billot et al., 2023; Guha Roy et al., 2019; Henschel et al., 2020, 2022; Estrada et al., 2023; Zhang et al., 2024). However, most existing approaches remain task-specific—trained for a single structure, cohort, or labeling rules—and demonstrate lower performance when deployed on heterogeneous datasets, limiting their generalization and clinical applicability.

To enhance flexibility and interpretability, large language models (LLMs) and vision-language models (VLMs) have been developed for medical image segmentation by coupling textual descriptions with visual representations (Ma et al., 2024; Fu et al., 2025; Su et al., 2025; Zhao et al., 2025). These multimodal models incorporate semantic context and enable adaptable segmentation across domains. Prompt-based VLMs extend this capability to open-vocabulary medical segmentation across organs and modalities (Ma et al., 2024; Zhao et al., 2025), while anatomical priors (e.g., shape templates and mesh constraints) further enhance spatial consistency (Su et al., 2025). In neuroimaging, emerging protocol-guided approaches encode hierarchical anatomical relationships, such as topology-based text generation to improve brain segmentation (Fu et al., 2025).

Despite these advances, subcortical segmentation still lacks clinically unified anatomical protocols and evaluation frameworks. First, current visual backbones are constrained by their training labels. Most large-scale datasets rely on FreeSurfer-derived masks because they are readily available (Fischl et al., 2002; Tae et al., 2008). Some models such as FastSurfer (Henschel et al., 2020, 2022; Estrada et al., 2023), SynthSeg (Billot et al., 2023), and QuickNAT (Guha Roy et al., 2019) largely reproduce or refine these outputs. However, FreeSurfer boundaries often diverge from expert manual labels (Morey et al., 2009; Schoemaker et al., 2016; Lerch et al., 2017), introducing systematic structural bias into both training and evaluation. Second, even when manual segmentations are available, there is still no clinically unified protocol: different experts and software tools apply different delineation rules—for example, outlining the amygdala or hippocampus with different boundaries—resulting in inconsistent ground-truth masks across datasets (Geuze et al., 2005; Yushkevich et al., 2015). Although recent vision–language models incorporate textual cues, they do not resolve this underlying protocol mismatch. Finally, current segmentation frameworks lack a standardized evaluation pipeline to assess anatomical accuracy from a

clinical perspective. Conventional metrics based on overlap, such as the Dice coefficient, quantify geometric similarity but fail to capture the morphological integrity, topological consistency, or biological validity of the predicted structures (Babalola et al., 2009).

To address these challenges, we propose **NeuroLangSeg**, a language-guided subcortical segmentation framework with pseudo-supervision and anatomical–linguistic validation based on a consistent anatomical protocol from Neuromorphometrics, Inc. (Landman and Warfield, 2012). Our main contributions are: 1) Contextual anatomical prompts are encoded and fused with visual features, enabling flexible, prompt-conditioned segmentation across structures and cohorts. 2) A unified visual backbone combines large-scale 3D masked autoencoder pretraining, label-efficient pseudo-label refinement, and global–local stabilization to improve robustness across scanners, ages, and modalities. 3) Clinical anatomical protocols encoded by an LLM guide morphological and topological discriminators. During inference, the evaluator integrates morphological, topological, and BrainChart-normalized volumetric metrics (adjusted for age and sex) to assess anatomical consistency.

Together, these components make NeuroLangSeg a clinically aligned and explainable framework for subcortical segmentation, providing both high accuracy and anatomical validation. To our knowledge, it is the first model to unify language-guided learning, semi-supervised segmentation, and anatomical–linguistic evaluation. Experiments on healthy and clinical cohorts show strong generalization and anatomically consistent performance across diverse populations.

## 2. Method

We address subcortical segmentation across heterogeneous MRI cohorts that differ in annotation policies and lack a unified, protocol-driven evaluation standard (Figure 1). Each sample contains a 3D MRI volume $X \in \mathbb{R}^{H \times W \times D}$ and its corresponding segmentation map $Y \in \mathbb{R}^{H \times W \times D}$. A textual prompt describing the target anatomical structure is provided and converted into a semantic embedding $T$, which is combined with visual features extracted from the MAE encoder to guide structure-specific prediction. The fused representation is passed to a segmentation decoder to produce structure-specific masks:

$$\hat{Y} = f(X, T) = \Psi(h_{\text{vis}}(g_{\text{vis}}(X)),\ h_{\text{query}}(g_{\text{vis}}(X),\ g_{\text{text}}(T))), \tag{1}$$

where $g_{\text{vis}}$ is the visual encoder, $h_{\text{vis}}$ is the visual decoder, $g_{\text{text}}$ is the text encoder, and $h_{\text{query}}$ is the query decoder. The segmentation head $\Psi$ performs dot-product matching and projection to generate the final mask $\hat{Y}$. Predictions are evaluated using morphological, topological, and volumetric metrics to assess anatomical consistency.

### 2.1. Language-Guided Prompt Encoding

**Text Encoder:** To encode anatomical concepts and their associated positional knowledge, we adopt a BERT-based text encoder $g_{text}$ initialized from a biomedical language model (Zhao et al., 2025) and further adapted through supervised fine-tuning. The encoder maps heterogeneous textual descriptions, including structure names, morphological definitions, and pairwise spatial relations into a unified embedding space. This allows the resulting text embedding $g_{\text{text}}(T)$ to capture structure-specific location cues and facilitates grounding of anatomical terms within the volumetric imaging space.

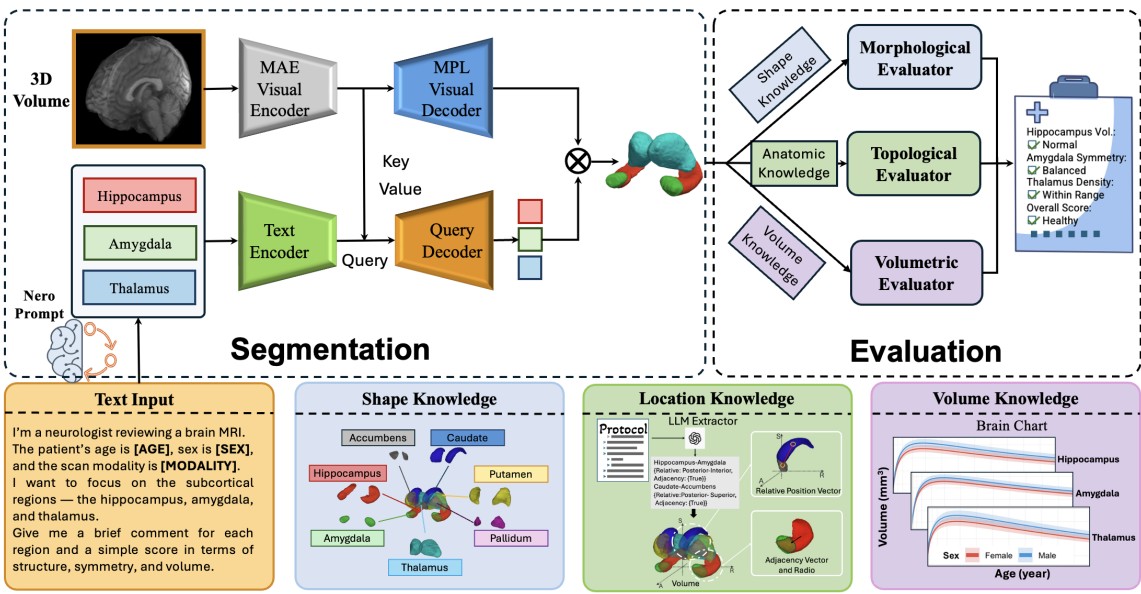

Figure 1: Overview of NeuroLangSeg Segmentation and Evaluation

**Query Decoder:** To adapt the text-derived representation to each MRI volume, we employ a Transformer-based query decoder $h_{\text{query}}$ that fuses textual embeddings with multi-scale visual features. The text embedding acts as the query, and image features serve as keys and values. A stack of cross-attention decoder blocks refines the query by attending to anatomy-relevant visual cues, enabling inference of subject-specific variations. The query $h_{\text{query}}(g_{\text{vis}}(X), g_{\text{text}}(T))$ is matched with voxel-level features $h_{\text{vis}}(g_{\text{vis}}(X))$ in the segmentation head, ensuring alignment between textual priors and the spatial context of the MRI scans.

## 2.2. Unified Visual Backbone for Label-Efficient Segmentation

We construct a unified visual backbone by combining 3D MAE pretraining, masked pseudo-label refinement, and global–local stabilization. This backbone provides a strong initialization for subsequent vision–language fine-tuning.

**MAE pretraining.** A 3D Masked Autoencoder (MAE) (He et al., 2021) is first trained on large-scale MRI volumes to learn modality- and site-invariant features. Local patches and a downsampled global view are randomly masked and reconstructed using an MSE loss, yielding a pretrained visual encoder $g_{\text{vis}}$.

**Masked Pseudo-Labeling (MPL).** To enable label-efficient domain adaptation, we adopt a 3D MPL teacher–student framework (Grill et al., 2020; Tarvainen and Valpola, 2017). We keep pretrained MAE encoder $g_{\text{vis}}$ with a segmentation decoder $h_{\text{vis}}$ to build segmentation $f_{\text{vis}} = h_{\text{vis}} \circ g_{\text{vis}}$. Given an input image $x_s$ and label $y_s$ from the source domain, the teacher model $f_\theta$ provides pseudo-labels for unlabeled target image $x_t$ and student model $f_\phi$ learns from masked source image $x_s^M$ and masked target image $x_t^M$ by minimizing the loss with weight $\beta$:

$$\mathcal{L}_{\text{MPL}} = \mathcal{L}_{\text{Seg}}\big(f_\phi(x_t^M),\, f_\theta(x_t)\big) + \beta\,\mathcal{L}_{\text{Seg}}\big(f_\phi(x_s^M),\, y_s\big). \tag{2}$$

Where $\mathcal{L}_{\text{Seg}}$ is a compound segmentation loss that consists of cross-entropy and Dice loss (Zhang et al., 2024).

**Global–Local Collaboration (GLC).** To stabilize pseudo-labels under domain shift, the GLC module (Zhang et al., 2024) fuses high-resolution local patches with global context extracted from the MAE encoder and regularizes their consistency. The full GLC formulation is provided in Appendix A. The visual backbone is pretrained with:

$$\mathcal{L}_{\text{vis}} = \mathcal{L}_{\text{FSS}} + \mathcal{L}_{\text{MPL}} + \mathcal{L}_{\text{GLC}}, \tag{3}$$

where $\mathcal{L}_{\text{FSS}} = \beta \, \mathcal{L}_{\text{Seg}}(f_\phi(x_s), y_s)$ is the loss of regular fully-supervised segmentation in source data and $\mathcal{L}_{\text{GLC}}$ contains the global–local consistency terms (Appendix A). After these stages, the visual backbone is fine-tuned jointly with the language-guided module using **only the supervised segmentation loss** $\mathcal{L}_{\text{FSS}}$.

### 2.3. Anatomical–Linguistic Discriminator

#### 2.3.1. MORPHOLOGICAL DISCRIMINATOR

Different subcortical structures exhibit distinct morphological variations, which serve as crucial reference points during manual annotation. Considering the shape characteristics of brain regions, the shape encoder $\mathcal{F}_{\text{shape}}$ employs an SE(3)-equivariant convolutional neural network (Billot et al., 2024) to extract shape features invariant to rigid transformations, mapping 3D annotations into a compact shape embedding space.

The shape encoder is pretrained using a denoising autoencoder framework, mapping noisy inputs to embeddings, which are reconstructed by a decoder comprising transposed 3D convolutions with instance normalization. The reconstruction loss combines MSE and soft Dice loss. During the training of NeuroLangSeg, the pre-trained shape encoder is used to constrain the morphological features. In each training step, both the prediction $\hat{Y}$ and the ground truth $Y$ are forwarded through the fixed $\mathcal{F}_{\text{shape}}$ to obtain their respective shape embeddings. The discrepancy between two embeddings is quantified using the MSE loss, which enforces the network to capture anatomically plausible shapes:

$$\mathcal{L}_{\text{shape}}(\hat{Y}, Y) = \left\| \mathcal{F}_{\text{shape}}\left(\hat{Y}\right) - \mathcal{F}_{\text{shape}}(Y) \right\|_2^2 \tag{4}$$

#### 2.3.2. TOPOLOGICAL DISCRIMINATOR

In addition to shape characteristics, the spatial relationships among subcortical nuclei provide crucial cues for manual annotation. To extract these positional features, we used a LLM to parse natural-language descriptions in annotation protocols provided by Neuromorphometrics, Inc. We used the following prompt to extract anatomical rules into a JSON format: *"Please extract the morphological features, relevant reference regions for manual annotation, and positional relationship descriptions... and convert them into a structured JSON description."* The LLM output identified 37 key anatomical pairs (15 left, 15 right, 7 cross-hemisphere) (shown in Appendix Table 4) and defined their relational types in a structured JSON format. For example, from the sentence *"the hippocampus is posterior and inferior to the amygdala,"* the LLM outputs structured JSON: *hippocampus-amygdala: {relative_position: [-1, -1, 0], adjacency_ratio: 1, adjacency_vector: [1, 1, 0]}*. The discrete

direction vector encodes the posterior–inferior offset under a standardized anatomical coordinate system (anterior, superior, right as positive). The adjacency ratio and vector denote whether two structures share a boundary and the dominant direction from one centroid toward the shared interface.

While the LLM identifies which relationships matter, the quantitative features are formalized by computing the statistics from the training set's ground truths. Each structure pair $(i, j)$ is thus represented by a 7D relational feature $\mathbf{r}_{ij} = [\Delta\mathbf{c}_{ij}, A_{ij}, \mathbf{d}_{ij}]$, where $\Delta\mathbf{c}_{ij}$ is the continuous relative position, $A_{ij}$ is the adjacency ratio, and $\mathbf{d}_{ij}$ is the adjacency-direction vector. To account for inter-subject variability in age and development, all relative position vectors are explicitly normalized based on the subject's total brain volume before being processed by the discriminator. For each subject, anatomical pairs form the relational matrix $\mathbf{R} \in \mathbb{R}^{K \times 7}$, encoding the full anatomical topology. $K = 37$ is the number of anatomical pairs. The MLP-based location encoder $\mathcal{F}_{\text{loc}}$ is pretrained in the task of reconstructing relative vectors $R$ extracted from annotation images of all subjects in the normal cohort. In NeuroLangSeg training, this fixed $\mathcal{F}_{\text{loc}}$ enforces topological consistency: for $\hat{Y}$ and $Y$, their relational matrices $\mathbf{R}_{\hat{Y}}$ and $\mathbf{R}_Y$ are extracted and encoded as global location embeddings. A Mean Squared Error (MSE) loss minimizes the discrepancy between the two embeddings, constraining the network to preserve accurate anatomical relationships:

$$\mathcal{L}_{\text{loc}}(\hat{Y}, Y) = \left\| \mathcal{F}_{\text{loc}}\left(\mathbf{R}_{\hat{Y}}\right) - \mathcal{F}_{\text{loc}}\left(\mathbf{R}_Y\right) \right\|_2^2 \tag{5}$$

### 2.4. Total Loss

The total training objective of **NeuroLangSeg** integrates supervised segmentation with protocol-guided anatomical constraints. The supervised term, $\mathcal{L}_{\text{FSS}}$, combines binary cross-entropy and soft Dice losses to encourage both voxel-level accuracy and region-level overlap fidelity. Two auxiliary regularizers are used: a shape loss $\mathcal{L}_{\text{shape}}$ that penalizes deviations from protocol-defined morphological characteristics, and a location loss $\mathcal{L}_{\text{loc}}$ that constrains predictions to anatomically valid spatial neighborhoods derived from protocol-based adjacency rules. The anatomical–linguistic discriminators that define these protocol constraints are not optimized jointly with the segmentation model; they are trained once using manual labels and a fixed anatomical protocol and are frozen during segmentation training and evaluation. The overall loss is defined as:

$$\mathcal{L}_{\text{total}}(\hat{Y}, Y) = \lambda_1 \mathcal{L}_{\text{FSS}}(\hat{Y}, Y) + \lambda_2 \mathcal{L}_{\text{shape}}(\hat{Y}, Y) + \lambda_3 \mathcal{L}_{\text{loc}}(\hat{Y}, Y), \tag{6}$$

where $\lambda_1$, $\lambda_2$, and $\lambda_3$ control the relative contributions of segmentation fidelity, morphological regularization, and anatomical location consistency.

## 3. Evaluation

### 3.1. Classical Metrics

We evaluate segmentation quality using Dice Similarity Coefficient (**DSC**) and Normalized Surface Distance (**NSD**) (Nikolov et al., 2021) against manual labels when available.

$$\text{DSC}(\hat{Y}, Y) = \frac{2|\hat{Y} \cap Y|}{|\hat{Y}| + |Y|}, \qquad \text{NSD}(\hat{Y}, Y) = \frac{|\partial\hat{Y} \cap B_{\partial Y}| + |\partial Y \cap B_{\partial\hat{Y}}|}{|\partial\hat{Y}| + |\partial Y|} \tag{7}$$

The **DSC** measures volumetric overlap between a prediction $\hat{Y}$ and the manual ground truth $Y$, and **NSD** evaluates boundary agreement within a tolerance $\tau$. $B_{\partial\hat{Y}}$ and $B_{\partial Y}$ denote tolerance bands around the prediction and ground-truth boundaries with $\tau = 1$.

### 3.2. Anatomical–Linguistic Evaluators

For large-scale or clinical datasets without manual annotations, we rely on three anatomical evaluators—morphological, topological, and volumetric.

The morphological evaluator assesses whether a predicted structure conforms to its anatomical shape. Using $\mathcal{F}_{\text{shape}}$, we derive per-label shape priors by encoding the annotations of all healthy instances of each structure and averaging them into a prototype vector $\boldsymbol{\mu}_{\text{shape}} \in \mathbb{R}^{128}$. During evaluation, $\hat{Y}$ is encoded by $\mathcal{F}_{\text{shape}}$, and its embedding is compared with the prototype via cosine similarity. This similarity is reported as the shape-consistency score, reflecting the morphological correctness of the prediction.

$$\text{Score}_{\text{shape}}(\hat{Y}) = \frac{\langle \mathcal{F}_{\text{shape}}(\hat{Y}), \boldsymbol{\mu}_{\text{shape}} \rangle}{\|\mathcal{F}_{\text{shape}}(\hat{Y})\|_2 \, \|\boldsymbol{\mu}_{\text{shape}}\|_2}, \tag{8}$$

The topology evaluator measures whether predicted regions preserve correct anatomical spatial relationships. We extracted the mean $\boldsymbol{\mu}_{\text{loc}} \in \mathbb{R}^{K \times 7}$ and standard $\boldsymbol{\sigma}_{\text{loc}} \in \mathbb{R}^{K \times 7}$ deviation of relational features across all annotated subjects. $K = 37$ is the number of anatomical pairs. During evaluation, relational features $\hat{\mathbf{R}} \in \mathbb{R}^{K \times 7}$ are extracted from the segmentation $\hat{Y}$, normalized using $\boldsymbol{\mu}_{\text{loc}}$ and $\boldsymbol{\sigma}_{\text{loc}}$, and converted into a topological correctness score:

$$\text{Score}_{\text{loc}}(\hat{\mathbf{R}}) = \exp\left(-\rho \frac{\sqrt{\sum (\hat{\mathbf{R}} - \boldsymbol{\mu}_{\text{loc}})^2 / \boldsymbol{\sigma}_{\text{loc}}^2}}{\sqrt{K}}\right), \tag{9}$$

The volumetric evaluator checks whether predicted structure volumes align with population norms. For each structure, the predicted volume $V$ is converted to an age- and sex-adjusted BrainChart $z$-score (Bethlehem et al., 2022; Rutherford et al., 2022), which we denote as $\text{Score}_{\text{vol}}$. Volumes with $|\text{Score}_{\text{vol}}| \leq 2$ are considered plausible:

$$\text{Score}_{\text{vol}} = \frac{V - \mu_{\text{ref}}(age, sex)}{\sigma_{\text{ref}}(age, sex)}. \tag{10}$$

## 4. Experiments

We evaluate NeuroLangSeg across three complementary settings: (1) in-site, (2) cross-site segmentation and generalization, and (3) clinical disease-cohort assessment. Segmentation accuracy (DSC, NSD) is reported wherever manual labels are available, while the three anatomical–linguistic evaluators (morphological, topological, volumetric) quantify anatomical robustness in both labeled and unlabeled datasets. Across all experiments, we compare NeuroLangSeg with four visual-only segmentation models (FastSurfer (Henschel et al., 2020, 2022), QuickNAT (Guha Roy et al., 2019), MAPSeg (Zhang et al., 2024), and nnU-Net (Isensee et al., 2021), as well as SAT (Zhao et al., 2025) as the vision-language baseline. FastSurfer (Henschel et al., 2022) baseline utilizes the latest VINNA architecture, which

incorporates an internal augmentation strategy for resolution independence. Notably, nnU-Net and MAPSeg serve as the underlying backbones for both SAT and NeuroLangSeg to ensure a controlled comparison of linguistic integration. While methods like SynthSeg (Billot et al., 2023) are popular for domain-agnostic full-brain segmentation, they were excluded here as they rely on intensity simulations for whole-brain labels and are not directly applicable to our focus on protocol-specific subcortical structures and anatomical-linguistic alignment.

## 4.1. Dataset

**MAE Pretraining:** We compile 11,948 unlabeled T1/T2 MRI scans spanning ages 1–100 years from nine publicly available datasets (e.g., **ABCD** (Casey et al., 2018) and **HCP** (Harms et al., 2018); full list in Appendix B). These scans contain no manual labels and are used solely for self-supervised MAE pretraining. **Pseudo-supervised Fine-tuning:** A total of 118 manually labeled T1-weighted subjects spanning ages 1–100 years are drawn from **ADNI** (Jack et al., 2008), **CANDI** (Kennedy et al., 2012), **OASIS** (Marcus et al., 2010), **Colin** (Holmes et al., 1998), and **BCP** (Howell et al., 2019) dataset. These subjects provide ground-truth annotations for supervised fine-tuning and in-site/cross-site segmentation evaluation. **Clinical Cohorts:** We additionally include two non–manually labeled clinical datasets—20 subjects from **BrainTS (BraTS)** (Li et al., 2023) tumor cohort and 30 subjects from **ADNI** (Jack et al., 2008) Alzheimer's disease cohort—which are used exclusively to evaluate out-of-distribution anatomical generalization without manual ground truth.

## 4.2. Experimental Settings

**In-Site Segmentation (Exp. 1):** We evaluate performance under matched training and testing conditions using the 118 manually labeled subjects. The dataset is randomly split 50% for training, 10% for validation, and 40% for testing.

**Cross-Site Generalization (Exp. 2):** To quantify generalization under realistic domain shift, we use the **ADNI** and **Colin** datasets as an external test cohort. 12 **ADNI** subjects and one **Colin** subject are withheld from finetuning, thereby providing an independent evaluation of cross-site performance.

**Disease–Cohort Assessment (Exp. 3):** To evaluate clinical robustness and out-of-distribution behavior, we apply NeuroLangSeg to the **BraTS** tumor dataset and the **ADNI** Alzheimer's cohort. Since no manual labels are available, evaluation is performed using the morphological, topological, and volumetric anatomical–linguistic evaluators.

## 4.3. Results and Discussion

### 1. Segmentation Accuracy and Visualization:

Table 1 summarizes the in/cross-site DSC/NSD performance. In in-site evaluation, NeuroLangSeg obtains the highest average DSC (86.9%) and NSD (95.0%), exceeding the strongest baseline by +1.9% DSC and +2.2% NSD, with larger gains over the average baseline (+4.1% DSC, +8.0% NSD), particularly for small structures such as the amygdala and accumbens. Under cross-site evaluation, NeuroLangSeg again achieves the highest

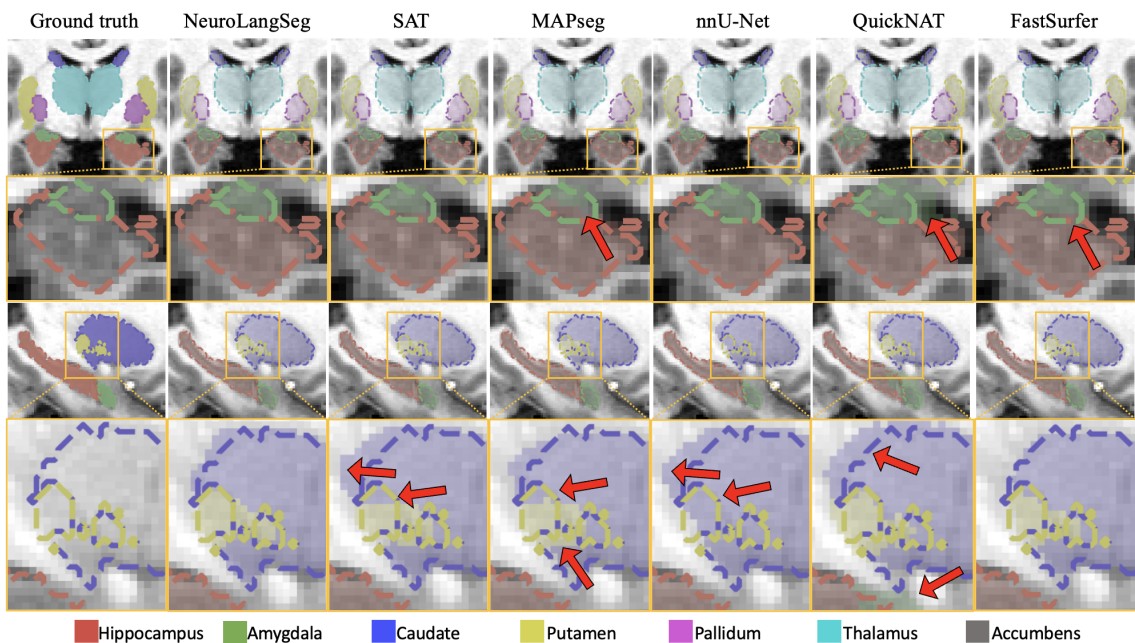

Figure 2: Qualitative comparisons. Coronal and sagittal planes, and zoomed-in regions of interest, respectively. Major segmentation errors are highlighted with red arrows. Ground-truth boundaries are indicated by dotted lines, while segmentations from different methods are shown as transparent overlays.

Table 1: Segmentation performance (DSC and NSD) across seven subcortical structures for in-site and cross-site generalization. Bold indicates the best performance.

| Method | DSC % | | | | | | | | NSD % | | | | | | | |
|---|---|---|---|---|---|---|---|---|---|---|---|---|---|---|---|---|
| | HIPP | AMG | CD | PT | PD | TM | AB | Avg | HIPP | AMG | CD | PT | PD | TM | AB | Avg |
| **In-site** | | | | | | | | | | | | | | | | |
| FastSurfer | 85.4 | 80.9 | 88.2 | 88.7 | 82.2 | 91.6 | 78.3 | 85.0 | 91.6 | 88.2 | 94.4 | 92.9 | 87.6 | 89.0 | 92.5 | 90.9 |
| QuickNAT | 77.5 | 59.5 | 80.7 | 83.4 | 72.7 | 87.6 | 59.3 | 74.4 | 78.2 | 43.2 | 79.4 | 82.5 | 61.0 | 74.6 | 61.9 | 68.7 |
| nnU-Net | 85.4 | 81.1 | 87.9 | 88.4 | 84.0 | 91.1 | 75.9 | 84.8 | 93.4 | 91.7 | 94.7 | 91.9 | 91.5 | 90.1 | 91.5 | 92.1 |
| MAPSeg | 85.7 | 79.8 | 88.3 | 88.7 | 84.5 | 91.5 | 76.8 | 85.0 | 91.9 | 85.5 | 95.0 | 93.2 | 88.6 | 88.8 | 91.3 | 90.6 |
| SAT | 85.7 | 81.2 | 87.4 | 88.3 | 84.4 | 90.9 | 77.1 | 85.0 | 92.6 | 91.7 | 95.2 | 95.1 | 91.3 | 89.6 | 93.8 | 92.8 |
| NeuroLangSeg | **87.6** | **84.0** | **88.9** | **89.3** | **86.2** | **92.0** | **80.4** | **86.9** | **95.3** | **94.9** | **97.2** | **96.1** | **93.5** | **92.8** | **95.7** | **95.0** |
| **Cross-site** | | | | | | | | | | | | | | | | |
| FastSurfer | 77.2 | 69.1 | 85.4 | 86.2 | 75.5 | 87.8 | 70.1 | 78.8 | 67.2 | 62.1 | 76.4 | 72.7 | 65.1 | 63.7 | 69.6 | 68.1 |
| QuickNAT | 74.2 | 63.1 | 76.8 | 80.5 | 66.5 | 86.6 | 64.0 | 73.1 | 61.1 | 42.6 | 60.2 | 59.7 | 37.1 | 57.8 | 53.6 | 53.2 |
| nnU-Net | **84.6** | 79.3 | **87.7** | 86.9 | 80.5 | 89.4 | 76.3 | 83.5 | 92.8 | 90.3 | 95.1 | 88.9 | 88.0 | 86.4 | 93.2 | 90.7 |
| MAPSeg | 83.9 | 79.3 | 87.3 | 87.9 | **81.6** | **89.8** | 77.0 | 83.8 | 90.3 | 87.8 | 95.9 | 95.1 | 89.7 | 87.1 | 92.1 | 91.1 |
| SAT | 83.2 | 77.6 | 86.7 | 86.7 | 80.1 | 89.0 | 75.3 | 82.6 | 91.5 | 90.2 | 96.4 | 95.2 | 89.8 | 87.5 | 94.4 | 92.2 |
| NeuroLangSeg | 84.5 | **80.0** | 86.8 | **88.1** | 81.0 | **89.8** | 78.1 | **84.0** | **93.5** | **92.8** | **97.1** | **97.0** | 91.5 | **89.7** | **96.0** | **93.6** |

HIPP:Hippocampus, AMG:Amygdala, TM:Thalamus, CD:Caudate, PT:Putamen, PD:Pallidum, AB:Accumbens

average DSC (84.0%) and NSD (93.6%). This yields +0.2% DSC and +1.4% NSD gain

Table 2: Segmentation performance (DSC and NSD) across seven subcortical structures for ablation study. Bold indicates the best performance.

| Method | DSC % | | | | | | | | NSD % | | | | | | | |
|---|---|---|---|---|---|---|---|---|---|---|---|---|---|---|---|---|
| | HIPP | AMG | CD | PT | PD | TM | AB | Avg | HIPP | AMG | CD | PT | PD | TM | AB | Avg |
| **In-site** | | | | | | | | | | | | | | | | |
| w/o discriminators | 86.2 | 81.9 | 87.5 | 88.0 | 84.5 | 91.0 | 77.5 | 85.2 | 93.8 | 92.8 | 95.9 | 94.8 | 91.2 | 89.8 | 94.0 | 93.2 |
| w/o morphological discriminator | 86.6 | 82.4 | 87.9 | 88.4 | 84.9 | 91.3 | 78.3 | 85.7 | 93.5 | 94.4 | 95.0 | 94.7 | 91.6 | 92.2 | 95.3 | 93.8 |
| w/o topological discriminator | 87.2 | 83.5 | 88.4 | 88.9 | 85.9 | 91.7 | 79.4 | 86.4 | 92.8 | **95.1** | 95.2 | 94.6 | **94.6** | **92.8** | 95.6 | 94.4 |
| NeuroLangSeg | **87.6** | **84.0** | **88.9** | **89.3** | **86.2** | **92.0** | **80.4** | **86.9** | **95.3** | 94.9 | **97.2** | **96.1** | 93.5 | **92.8** | **95.7** | **95.0** |
| **Cross-site** | | | | | | | | | | | | | | | | |
| w/o discriminators | 84.1 | 79.3 | 86.5 | 87.5 | 80.1 | 89.3 | 77.5 | 83.5 | 92.6 | 91.9 | 96.7 | 96.6 | 91.3 | 89.0 | 95.7 | 93.4 |
| w/o morphological discriminator | 84.3 | 79.6 | 86.5 | 87.7 | 80.1 | 89.4 | 77.2 | 83.5 | 92.8 | 92.4 | 96.6 | 96.8 | 91.4 | 89.2 | 95.3 | 93.5 |
| w/o topological discriminator | **84.9** | 79.7 | 86.7 | 88.0 | 80.3 | 89.8 | 77.5 | 83.8 | **93.8** | **93.3** | **97.2** | **97.3** | 91.3 | **90.8** | 95.9 | **94.2** |
| NeuroLangSeg | 84.5 | **80.0** | 86.8 | 88.1 | 81.0 | 89.8 | **78.1** | **84.0** | 93.5 | 92.8 | 97.1 | 97.0 | **91.5** | 89.7 | **96.0** | 93.6 |

HIPP:Hippocampus, AMG:Amygdala, TM:Thalamus, CD:Caudate, PT:Putamen, PD:Pallidum, AB:Accumbens

Table 3: Anatomical–Linguistic Evaluators' average scores for in-site, cross-site (CN), and clinical cohorts (AD and tumor). *** indicate statistically significant differences with $p < 0.001$. Stable CN scores indicate protocol-consistent anatomy, while significant deviations in AD and tumor reflect pathological changes.

| Score | In-site and Cross-site (CN) | | | | | | ADNI (AD) | BraTS (Tumor) |
|---|---|---|---|---|---|---|---|---|
| | FastSurfer | QuickNAT | nnU-Net | MAPSeg | SAT | NeuroLangSeg | NeuroLangSeg | NeuroLangSeg |
| Shape | $78.1 \pm 1.7$ | $78.9 \pm 1.9$ | $79.4 \pm 1.3$ | $79.0 \pm 1.3$ | $79.5 \pm 1.4$ | $79.2 \pm 1.7$ | $71.2 \pm 1.4^{***}$ | $71.4 \pm 1.4^{***}$ |
| Location | $87.9 \pm 2.8$ | $84.0 \pm 2.5^{***}$ | $86.9 \pm 4.2$ | $88.3 \pm 2.6$ | $87.3 \pm 2.9$ | $87.9 \pm 2.4$ | $77.1 \pm 2.7^{***}$ | $67.4 \pm 3.5^{***}$ |
| Volume | $1.39 \pm 0.51^{***}$ | $2.17 \pm 0.32^{***}$ | $1.05 \pm 0.34$ | $1.02 \pm 0.35$ | $1.09 \pm 0.35$ | $0.94 \pm 0.24$ | $1.82 \pm 0.49^{***}$ | $2.24 \pm 0.53^{***}$ |

over the strongest baseline and substantial improvements over the average baseline (+3.6% DSC, +14.5% NSD).

Figure 2 shows in-site qualitative segmentation results. In the coronal view, several baselines enlarge or shrink the amygdala. In the sagittal view, manual annotations are inherently discontinuous because they were drawn primarily in the coronal plane. Models trained only on these labels tend to compensate incorrectly: methods such as SAT, MAPSeg, and QuickNAT often enlarge the structure, FastSurfer tends to shrink it, and nnU-Net frequently yields missing segments. In contrast, NeuroLangSeg produces anatomically consistent shapes across views without artificial expansion, collapse, or disappearance. Minor 1–2 pixel over-segmentation may still occur, which is a common and well-known behavior in deep learning–based segmentation methods.

Table 2 reports ablation results on the in-site dataset. Removing all discriminators leads to a clear performance degradation in both Dice and NSD across all subcortical structures. Excluding either the shape or location discriminator generally reduces overall performance of segmentation accuracy compared with the full model.

## 2. Anatomical–Linguistic Evaluation and Clinical Generalization:

Table 3 reports the average anatomical–linguistic evaluator scores for in-site and cross-site cognitively normal (CN) subjects and clinical cohorts. For CN participants, NeuroLangSeg achieves high shape and location scores and volume scores close to 0 (within $[-2, 2]$), indicating good alignment with the anatomical protocol. Volume score is summarized using $|\text{Score}_{\text{vol}}|$ to properly capture deviation magnitude. Other segmentation

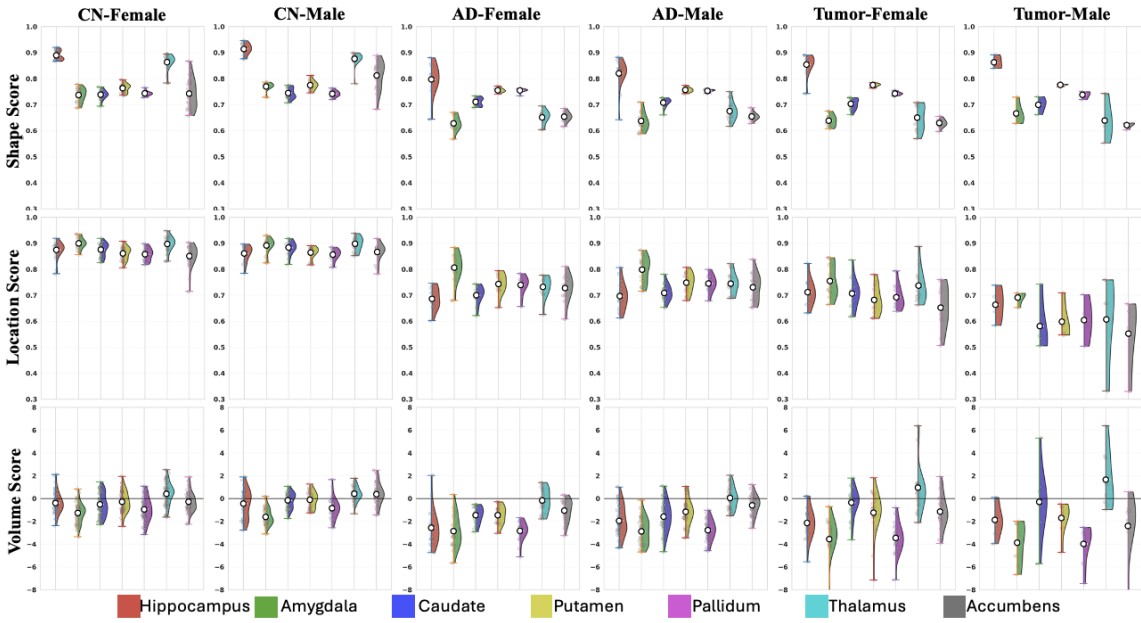

Figure 3: Shape, location, and volume-score distributions of subcortical regions in Cognitively Normal (CN), Alzheimer's Disease (AD), and tumor participants, stratified by sex.

methods also produce CN scores that cluster near the reference values, as expected for healthy controls; however, their morphological, topological, and volumetric metrics are consistently lower than those of NeuroLangSeg. As expected, ADNI (AD) and BraTS (tumor) cohorts show reduced scores due to pathology-related changes in morphology and spatial organization.

A one-way ANOVA was used to assess whether the evaluators distinguish anatomical quality. Within CN subjects, evaluator scores from each method were compared to NeuroLangSeg. Most methods showed no significant difference, as all were tested on the same CN cohort. Only QuickNAT and FastSurfer differed significantly ($p < 0.001$), consistent with their lower DSC/NSD performance. Using NeuroLangSeg across CN, AD, and tumor groups, all three evaluators showed significant differences ($p < 0.001$), indicating stable scores in healthy controls and clear sensitivity to disease-related anatomical changes.

Figure 3 illustrates these patterns using violin plots of our method's evaluator score distributions for CN, AD, and tumor participants. CN subjects cluster tightly around the reference values, whereas AD and tumor display volume outside $[-2, 2]$, and reductions in shape and location scores. Slightly lower CN scores for the amygdala and pallidum arise because our manual labels are smaller than the FreeSurfer-derived volumes used in the BrainChart reference.

## 5. Conclusion

We introduced **NeuroLangSeg**, a language-guided framework that unifies visual features with protocol-consistent anatomical reasoning for subcortical MRI segmentation. Through MAE pretraining, pseudo-supervised fine-tuning, and anatomical–linguistic evaluation, our method delivers accurate, consistent, and clinically interpretable segmentations across in-site, cross-site, and disease cohorts. Quantitative comparison with state-of-the-art models shows substantial gains, including **+8.0 NSD** in in-site evaluation and **+14.5 NSD** in cross-site generalization over the average baseline. ANOVA analyses further confirm that our anatomical–linguistic scores significantly distinguish healthy controls from pathological cases while remaining stable within CN subjects. By grounding segmentation in a standardized anatomical protocol, NeuroLangSeg advances robust, interpretable, and clinically aligned neuroimaging segmentation. The future work is to extend NeuroLangSeg to infant, pediatric, and fetal MRI, as well as to additional disease cohorts, to further assess robustness across developmental stages and pathological conditions.

## Acknowledgments

This work was supported by NIH grants R00HD103912, R01MH133313 (Y.W.), and Neuromorphometrics, Inc.

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

## Appendix A. Model Architecture

### A.1. Text Encoder

Using text encoder $g_{\text{text}}$, each text prompt $T$ including structure names, morphological definitions, and pairwise spatial relations is tokenized and processed through a multi-layer self-attention stack, producing a d-dimensional representation. Given a batch of paired textual descriptions $(T_i, T_i')$, each anatomical structure's concept $T_i$ is paired with $T_i'$, corresponding either to (1) a descriptive phrase capturing its anatomical morphology $T_i^{\text{mor}}$, or (2) relational statements describing its spatial relations relative to other structures $T_i^{\text{rela}}$. The encoder produces embedding pairs $(z_i, z_i')$:

$$z_i = g_{\text{text}}(T_i), \qquad z_i' = g_{\text{text}}(T_i'), \qquad z_i, z_i' \in \mathbb{R}^d \tag{A1}$$

To encourage semantically aligned anatomical concepts to share a nearby representation, we optimized $g_{\text{text}}$ using contrastive learning, with an InfoNCE contrastive loss.

$$L_{\text{text}} = -\frac{1}{N} \sum_{i=1}^{N} \left[ \log \frac{\exp(\frac{z_i.z_i'}{\tau})}{\sum_{k=1}^{N} \mathbf{1}_{i \neq k} \exp(\frac{z_i.z_k'}{\tau})} + log \frac{\exp(\frac{z_i.z_i'}{\tau})}{\sum_{k=1}^{N} \mathbf{1}_{i \neq k} \exp(\frac{z_k.z_i'}{\tau})} \right] \tag{A2}$$

where $\tau$ is the temperature coefficient, and $N$ is the number of subcortical regions.

### A.2. Query Decoder

The text embedding $z$ serves as the initial *Query*, while the visual feature $g_{vis}(X)$ extracted from the image encoder serves as the *Keys* and *Values*. A stack of decoder blocks, each consisting of cross-attention followed by feed-forward layers, progressively refines the representation:

$$q = h_{\text{query}}(g_{\text{vis}}(X),\, g_{\text{text}}(T))\,, \qquad q \in \mathbb{R}^d \tag{A3}$$

Through this cross-attention mechanism, the query decoder allows the text embedding to attend to anatomically relevant visual cues, enabling the model to infer subject-specific variations in location, orientation, and shape. The output $q$ serves as an image-conditioned anatomical query, which is subsequently matched against voxel-wise visual features $h_{\text{vis}}(g_{\text{vis}}(X))$ in the segmentation head, using a dot-product operation to generate the final segmentation.

### A.3. 3D Multi-Scale Masked Autoencoder (MAE)

Our 3D MAE uses 3D ResNet blocks (Zhang et al., 2024) instead of Vision Transformers. The encoder is composed of eight 3D ResNet blocks and we adopt an asymmetric architecture with a lightweight decoder, detailed in Supplementary Figure 4. During training, the model jointly learns from two input types—randomly sampled local patches $x$ and a downsampled version of the full volumetric scan $X$, both resized to $96^3$ voxels. To enable self-supervised learning, both $x$ and $X$ are partitioned into non-overlapping 3D patches and subjected to random masking. For $x$, we use a patch size of $8^3$ and mask 70% of the patches uniformly at random. For $X$, which provides a broader field of view (FOV), we use

a smaller patch size of $4^3$ while maintaining the same masking ratio. The resulting masked inputs, denoted as $x_M$ and $X_M$, are passed to the MAE, which is trained to reconstruct the original unmasked volumes using mean squared error loss computed only on the masked regions.

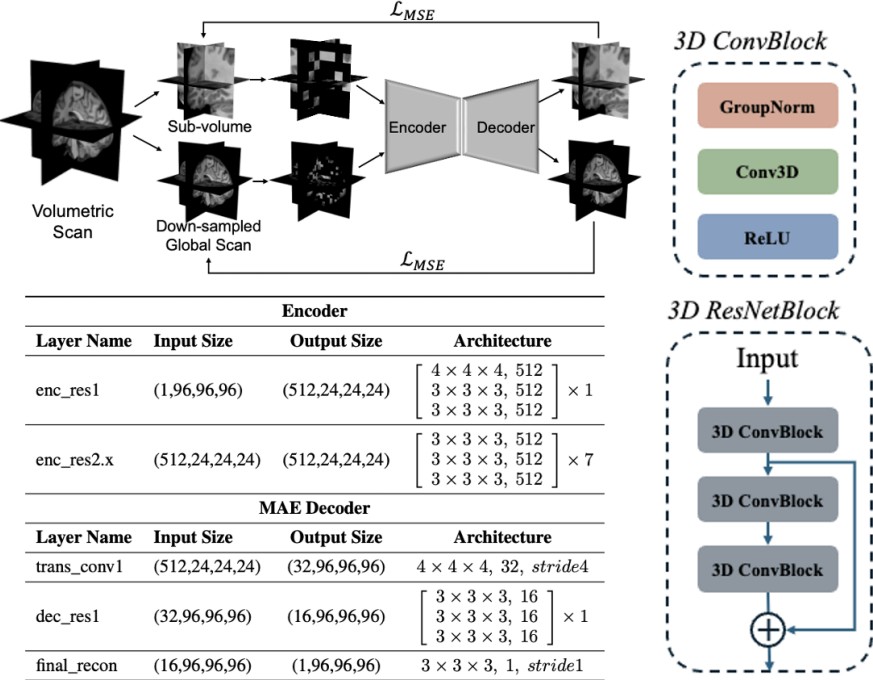

Figure 4: Illustrations of MAE 3D ResNet Block and 3D architectures.

## A.4. 3D Masked Pseudo-Labeling (MPL)

MAPSeg employs a 3D **Masked Pseudo-Labeling (MPL)** strategy based on a teacher–student architecture (Figure 5). The segmentation backbone combines the pretrained MAE encoder $g_{\mathrm{vis}}$ with a lightweight 3D decoder $h_{\mathrm{vis}}$ adapted from DeepLabV3. The decoder uses a 3D Atrous Spatial Pyramid Pooling (ASPP) module with multi-scale dilated convolutions to enlarge the effective receptive field. The student model processes both labeled source volumes and unlabeled target volumes, while the teacher model—an exponential moving average (EMA) of the student—produces stable pseudo-labels for the target domain.

To improve generalization, the student receives masked input volumes, following the same masking scheme used during MAE pretraining. This forces the model to rely on global context rather than local intensity alone. MPL integrates (i) supervised loss on source data and (ii) consistency loss between teacher and student predictions on target data. This yields a label-efficient adaptation mechanism that leverages MAE-learned priors while mitigating noisy pseudo-label propagation.

The teacher model's parameters $\theta$ are updated during training via an exponential moving average (EMA) of the student model's parameters $\phi$ (Tarvainen and Valpola, 2017):

$$\theta_{t+1} \leftarrow \alpha\theta_t + (1 - \alpha)\phi_t, \tag{A4}$$

where $t$ and $t + 1$ denote training iterations and $\alpha$ is the EMA update weight. For models initialized from large-scale MAE pretraining, we set $\alpha = 0.999$ during the first 1,000 steps and $\alpha = 0.9999$ afterwards. For models pretrained on small-scale source and target datasets (e.g., only dozens of scans), we set $\alpha = 0.99$ during the first 1,000 steps, $\alpha = 0.999$ during the next 2,000 steps, and $\alpha = 0.9999$ for the remaining training. The teacher model $f_\theta$ is initialized with the student model's parameters $\phi$ after a warm-up stage (e.g., 1,000 iterations) on the source-domain data.

### A.5. 3D Global-Local Collaboration (GLC)

To improve pseudo-label stability under large domain shifts, we introduce a **Global–Local Collaboration (GLC)** module (Zhang et al., 2024). For each scan (Figure 5), we extract a high-resolution local patch $x$ and a downsampled global volume $X$. The encoder $g_{\text{vis}}$ produces local and global features:

$$\chi_{\text{loc}} = g_{\text{vis}}(x), \qquad \chi_{\text{glo}} = \text{upsample}(M \odot g_{\text{vis}}(X)), \tag{A5}$$

where $M$ is a binary mask and $\odot$ denotes cropping followed by interpolation to match spatial dimensions. The GLC module fuses the two feature streams by channel-wise concatenation:

$$f_{\text{vis}}(x) = h_{\text{vis}}(\chi_{\text{loc}} \oplus \chi_{\text{glo}}), \tag{A6}$$

forming a unified 1024-dimensional latent representation processed by the ASPP head for segmentation. To provide global supervision, the model also predicts from the global view via

$$f_{\text{vis}}(X) = h_{\text{vis}}(g_{\text{vis}}(X) \oplus g_{\text{vis}}(X)). \tag{A7}$$

To enforce alignment between local and global information, we impose a cosine similarity regularizer:

$$\mathcal{L}_{\cos}(x, X) = 1 - \frac{\chi_{\text{loc}} \cdot \chi_{\text{glo}}}{\max(\|\chi_{\text{loc}}\|_2, \|\chi_{\text{glo}}\|_2, \epsilon)}. \tag{A8}$$

The GLC losses for source and target data are:

$$\mathcal{L}_{\text{GLC}}^S = \gamma\left[\mathcal{L}_{\text{Seg}}(f_\phi(X_s), Y_s) + \mathcal{L}_{\text{Seg}}(f_\phi(X_s^M), Y_s)\right] + \delta\left[\mathcal{L}_{\cos}(x_s, X_s) + \mathcal{L}_{\cos}(x_s^M, X_s^M)\right], \tag{A9}$$

$$\mathcal{L}_{\text{GLC}}^T = 2\gamma\,\mathcal{L}_{\text{Seg}}(f_\phi(X_t^M), f_\theta(X_t)) + 2\delta\,\mathcal{L}_{\cos}(x_t^M, X_t^M). \tag{A10}$$

During training, local $96 \times 96 \times 96$ patches are randomly sampled to provide high-resolution detail while the global branch maintains coarse contextual awareness. At inference, predictions are generated with a sliding-window scheme (stride 80) to cover the full volume.

With the supervised loss

$$\mathcal{L}_{\text{FSS}} = \beta\,\mathcal{L}_{\text{Seg}}(f_\phi(x_s), y_s), \tag{A11}$$

the full training objective becomes:

$$\mathcal{L}_{\text{vis}} = \mathcal{L}_{\text{FSS}} + \mathcal{L}_{\text{MPL}} + \mathcal{L}_{\text{GLC}}, \qquad \mathcal{L}_{\text{GLC}} = \mathcal{L}_{\text{GLC}}^S + \mathcal{L}_{\text{GLC}}^T. \tag{A12}$$

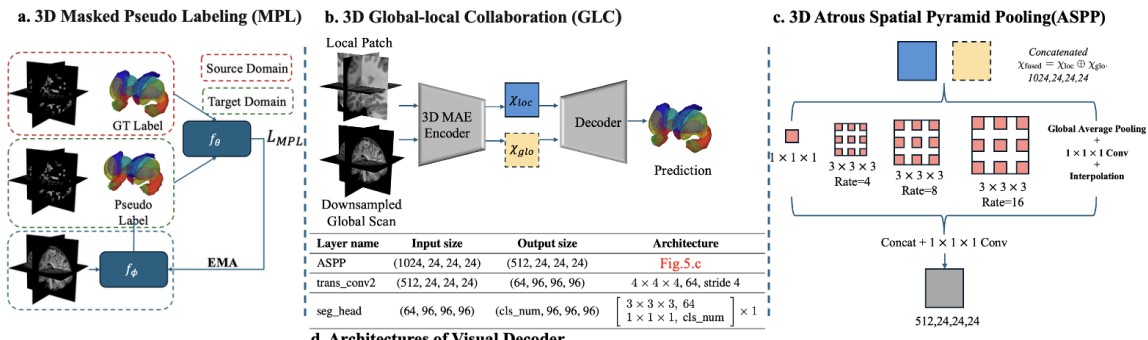

Figure 5: Illustrations of MPL and GLC.

## A.6. Morphological Discriminator

The morphological discriminator adopts the SE(3) -equivariant convolutional neural network (Billot et al., 2024) as the shape encoder $\mathcal{F}_{\text{shape}}$ to extract features. The shape encoder is pretrained using a denoising autoencoder framework, mapping noisy inputs to embeddings, which are reconstructed by a decoder comprising transposed 3D convolutions with instance normalization. For framework details, please refer to the supplementary Figure 6.

The reconstruction task is trained by minimizing the difference between the reconstructed image and the input image, and the loss uses a combination of multi-class Dice loss and MSE loss. For the input labeled image $Y$, the reconstructed labeled image $Y^{\text{recon}}$ is obtained after passing through the shape encoder and decoder. The loss is calculated as follows:

$$\mathcal{L}_{\text{shape\_recon}} = 1 - \frac{2|Y^{\text{recon}} \cap Y|}{|Y^{\text{recon}}| + |Y|} + \|Y^{\text{recon}} - Y\|_2^2 \tag{A13}$$

Figure 6: Illustrations of Morphological Discriminator.

## A.7. Topological Discriminator

Topological Discriminator employs MLP as the location encoder $\mathcal{F}_{\text{loc}}$, which is pretrained to reconstruct relation features from noisy inputs. For framework details, please refer to the supplementary Figure 7. For each subject, we extract relation vectors of 37 anatomical

pairs $\mathbf{R} \in \mathbb{R}^{K \times 7}$ from ground truth. $K = 37$ is the number of anatomical pairs. The relation feature $\mathbf{r}_{ij} = [\Delta\mathbf{c}_{ij}, A_{ij}, \mathbf{d}_{ij}]$ is composed of the continuous relative position $\Delta\mathbf{c}_{ij}$, the adjacency ratio $A_{ij}$, and the adjacency-direction $\mathbf{d}_{ij}$. $\Delta\mathbf{c}_{ij} = c_i - c_j$ is obtained by taking the difference between the centroids $c_i$ of structure $i$ and $c_j$ of structure $j$. Adjacency ratio $A_{ij} = \frac{\|N_{i \to j}\|}{\|B_i\|}$ is the proportion of shared boundary voxels, and the adjacency-direction vector $d_{ij} = c_i - \frac{\sum_{v \in N_{i \to j}} v}{\|N_{i \to j}\|}$ is defined as the difference between the centroid of the subset of structure $i$'s boundary voxels that are adjacent to $j$ and the centroid of structure $i$ as a whole. Here, $N_{i \to j}$ is the subset of boundary voxels of structure $i$ that are in direct spatial contact with structure $j$, and $B_i$ denotes the set of all boundary voxels belonging to anatomical structure $i$. Then we generate n = 100 perturbation versions in the relation vectors $R$ by adding Gaussian noise ($\sigma = 0.1$). The encoder maps noisy relations to embeddings, which are reconstructed by a feedforward decoder. The reconstruction loss is:

$$\mathcal{L}_{\text{loc\_recon}} = \|\mathbf{R} - \mathbf{R}^{\text{recon}}\|_2^2 \tag{A14}$$

where $\mathbf{R}^{\text{recon}}$ denotes the reconstructed relation matrix.

The details of the parameter settings for the pre-training tasks of the shape discriminator and the topology discriminator are presented in Table 11.

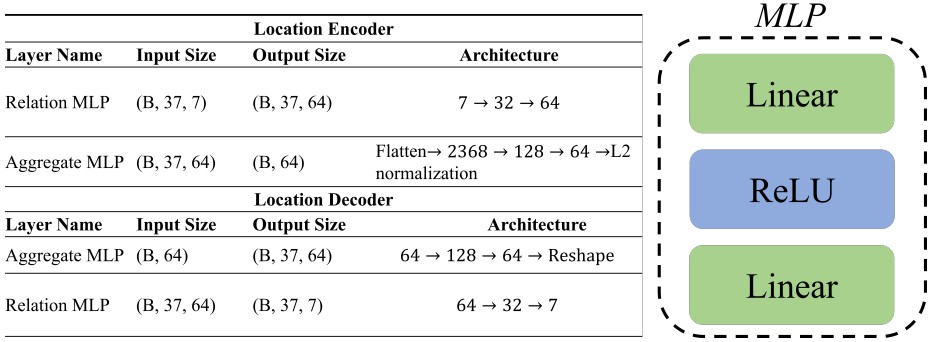

| Location Encoder | | | |
|---|---|---|---|
| **Layer Name** | **Input Size** | **Output Size** | **Architecture** |
| Relation MLP | (B, 37, 7) | (B, 37, 64) | 7 → 32 → 64 |
| Aggregate MLP | (B, 37, 64) | (B, 64) | Flatten→ 2368 → 128 → 64 →L2 normalization |
| Location Decoder | | | |
| **Layer Name** | **Input Size** | **Output Size** | **Architecture** |
| Aggregate MLP | (B, 64) | (B, 37, 64) | 64 → 128 → 64 → Reshape |
| Relation MLP | (B, 37, 64) | (B, 37, 7) | 64 → 32 → 7 |

Figure 7: Illustrations of Topological Discriminator.

## Appendix B. Dataset Description

We gather T1-weighted and T2-weighted MRI data across 1-100 years from 15 publicly available datasets. The detailed dataset information is as follows:

- **ABCD: Adolescent Brain Cognitive Development Study** (Casey et al., 2018) is a large-scale, longitudinal neuroimaging and behavioral study tracking brain development and child health in over 10,000 U.S. children aged 9–10 years. Participants were enrolled at ages 9–10 and are being followed into their early 20s. We collected 2930 subjects with 3211 longitudinal scans spanning 9-17 years, with 3211 T1-weighted and 3209 T2-weighted images.

- **ABIDE-I: Autism Brain Imaging Data Exchange** (Di Martino et al., 2014) is a cross-sectional multi-site initiative that shares resting-state fMRI and structural MRI

Table 4: List of the 37 anatomically relevant structure pairs used to construct relation vectors.

| Category | Structure pairs |
|---|---|
| Left intra-hemispheric (15) | (L-HIPP, L-AMG), (L-HIPP, L-TM), (L-HIPP, L-AB), |
| | (L-AMG, L-AB), (L-AMG, L-TM), (L-AMG, L-PT), |
| | (L-CD, L-PT), (L-CD, L-PD), (L-CD, L-AB), |
| | (L-CD, L-TM), (L-PT, L-PD), (L-PT, L-AB), |
| | (L-PT, L-TM), (L-PD, L-TM), (L-AB, L-TM) |
| Right intra-hemispheric (15) | (R-HIPP, R-AMG), (R-HIPP, R-TM), (R-HIPP, R-AB), |
| | (R-AMG, R-AB), (R-AMG, R-TM), (R-AMG, R-PT), |
| | (R-CD, R-PT), (R-CD, R-PD), (R-CD, R-AB), |
| | (R-CD, R-TM), (R-PT, R-PD), (R-PT, R-AB), |
| | (R-PT, R-TM), (R-PD, R-TM), (R-AB, R-TM) |
| Inter-hemispheric (7) | (L-HIPP, R-HIPP), (L-AMG, R-AMG), (L-CD, R-CD), |
| | (L-PT, R-PT), (L-PD, R-PD), (L-TM, R-TM), (L-AB, R-AB) |

data from individuals with autism and typically developing controls. We collected 1102 subjects/scans of T1-weighted images spanning 6 - 64 years.

- **ADHD-200: ADHD-200 Global Competition** (Bellec et al., 2017) is a cross-sectional multi-site dataset sharing resting-state fMRI and structural MRI data to identify biomarkers of Attention Deficit Hyperactivity Disorder (ADHD). We collected 869 subjects/scans of T1-weighted images spanning 7 - 26 years.

- **ADNI: Alzheimer's Disease Neuroimaging Initiative** (Jack et al., 2008) is a longitudinal, multi-site study designed to develop clinical, imaging, genetic, and biochemical biomarkers for early detection and tracking of Alzheimer's disease. We collected 50 subjects/scans of T1-weighted images spanning 60-96 years.

- **BCP: Baby Connectome Project** (Howell et al., 2019) is a longitudinal neuroimaging study aiming to map early brain development and connectivity from infancy through early childhood. We collected 2126 subjects with 2444 longitudinal scans spanning 0-7 years, with 2406 T1-weighted and 2347 T2-weighted images.

- **HBN: Healthy Brain Network** (Alexander et al., 2017) is a cross-sectional transdiagnostic pediatric study collecting neuroimaging, behavioral, cognitive, and genetic data to better understand mental health and learning disorders. We collected 1729 subjects/scans spanning 5 - 22 years, with 1698 T1-weighted and 562 T2-weighted images.

- **HCP-A: Human Connectome Project – Aging** (Bookheimer et al., 2019) is a cross-sectional dataset focused on understanding brain connectivity and aging across the adult lifespan. We collected 725 subjects/scans spanning 36 - 100 years, with 725 T1-weighted and 725 T2-weighted images.

- **HCP-D: Human Connectome Project – Development** (Somerville et al., 2018) is a cross-sectional study examining brain development and connectivity from child-

hood through young adulthood. We collected 652 subjects/scans spanning 6 - 22 years, with 652 T1-weighted and 652 T2-weighted images.

- **HCP-YA: Human Connectome Project – Development** (Harms et al., 2018) is a cross-sectional study to map the healthy human connectome by collecting and freely distributing neuroimaging and behavioral data on 1,200 normal young adults, aged 22-35.

- **PING: Pediatric Imaging, Neurocognition, and Genetics** (Jernigan et al., 2016) is a cross-sectional study designed to assess brain development and its genetic and environmental influences in children and adolescents. We collected 754 subjects/scans spanning 0 - 22 years, with 752 T1-weighted and 106 T2-weighted images.

- **CANDI: Child and Adolescent Neuro Development Initiative** (Kennedy et al., 2012) includes structural MRI scans of children and adolescents, supporting research on brain development, psychiatric disorders, and neuroanatomical differences across diagnoses.

- **OASIS: Open Access Series of Imaging Studies** (Marcus et al., 2010) provides structural brain MRI data across the adult lifespan, including individuals with and without Alzheimer's disease, to support neurodegenerative and aging research.

- **COLIN: Colin27 Brain Atlas** (Holmes et al., 1998) is a high-resolution MRI brain template created by averaging 27 T1-weighted scans of a single individual.

- **BraTS2023: The Brain Tumor Segmentation (BraTS) Challenge 2023** (Li et al., 2023) provides an expanded multi-site mpMRI dataset ( 4,500 cases) with expert tumor delineations across diverse populations and tumor types, enabling benchmarking of segmentation, missing-data handling, and cross-task generalizability.

We pretrain our models on a dataset of approximately 12,000 subjects, including both T1-weighted and T2-weighted scans. All datasets were preprocessed using N4 bias correction and skull stripping. Full dataset details are provided in Supplementary Table 5.

We use 118 manually labeled subjects from **ADNI** (Jack et al., 2008), **CANDI** (Kennedy et al., 2012), **OASIS** (Marcus et al., 2010), **Colin** (Holmes et al., 1998), and **BCP-50** (Howell et al., 2019) (details in Supplementary Table 6). For robust model development, 50% of subjects are used for training, 10% for validation, and 40% are held out for testing. In our study, **CANDI**, **Colin**, and **OASIS** are single-site datasets. The **ADNI** manual dataset includes 30 subjects spanning 25 distinct sites. **ADNI** subjects are split such that 15 subjects from 12 sites are used for training, 3 subjects from 2 sites are used for validation, and 12 subjects from 11 different sites are used for testing. **Colin** and 12 **ADNI** testing subjects are reserved exclusively for cross-site inference. Two non–manually labeled clinical datasets—**BraTS**(Li et al., 2023) tumor cohort and **ADNI** (Jack et al., 2008)(Alzheimer's disease) cohort (Table 7)—which are used exclusively to evaluate out-of-distribution anatomical generalization.

Table 5: Dataset summary across age ranges for pretraining: modality, number of subjects, scans, and age span.

| Dataset | Modality | Subjects | T1 Scans | T2 Scans | Age (yrs) |
|---------|----------|----------|----------|----------|-----------|
| **ABCD** | T1w, T2w | 2930 | 3211 | 3209 | 9–16 |
| **ABIDE-I** | T1w | 1102 | 1102 | – | 6–64 |
| **ADHD-200** | T1w | 869 | 869 | – | 7–26 |
| **BCP** | T1w, T2w | 2126 | 5183 | 4303 | 1–7 |
| **HBN** | T1w, T2w | 1729 | 1684 | 560 | 6–64 |
| **HCP-A** | T1w, T2w | 725 | 725 | 725 | 36–100 |
| **HCP-D** | T1w, T2w | 652 | 652 | 652 | 6–21 |
| **HCP-YA** | T1w, T2w | 1061 | 10 | 660 | 22–35 |
| **PING** | T1w, T2w | 754 | 752 | 106 | 3–21 |

Table 6: Dataset summary for subcortical segmentation: modality, number of subjects, scans, and age span.

| Dataset | Modality | Subjects | T1 Scans | T2 Scans | Age (yrs) |
|---------|----------|----------|----------|----------|-----------|
| **OASIS** | T1w | 50 | 70 | – | 18–93 |
| **ADNI** | T1w | 29 | 30 | – | 71-88 |
| **CANDI** | T1w | 13 | 13 | – | 5-15 |
| **Colin** | T1w | 1 | 1 | – | 27 |
| **BCP** | T1w, T2w | 25 | 25 | 25 | 1-2 |

## Appendix C. Baseline

### C.1. FastSurfer

FastSurferCNN (Henschel et al., 2020) is a 2D fully convolutional architecture designed for fast whole-brain segmentation and serves as the segmentation module within the Fast-Surfer pipeline. The network follows an encoder–decoder design similar to QuickNAT but introduces several architectural improvements: competitive dense blocks (replacing concatenation with maxout operations) to encourage feature competition, unpooling layers for spatially accurate upsampling, and a wider contextual field to better capture neuroanatomical boundaries. FastSurferCNN predicts 2D segmentations for axial, coronal, and sagittal slices, which are combined through a multi-view aggregation strategy to produce the final 3D mask.

In its original formulation, FastSurfer is trained on FreeSurfer-derived labels for 95 anatomical structures, providing a high-speed alternative to traditional surface-based processing. Because the method relies on 2D slice-wise predictions, its performance can vary across views, especially for small or discontinuous subcortical structures.

For our study, we employ FastSurferCNN as a baseline and fine-tune it on our 7-class subcortical label set under the same training conditions as the other methods.

Table 7: Dataset summary for clinical generalization: modality, number of subjects, scans, and age span.

| Dataset | Modality | Subjects | T1 Scans | T2 Scans | Age (yrs) |
|---------|----------|----------|----------|----------|-----------|
| **ADNI** | T1w | 30 | 30 | – | 60-96 |
| **BraTS** | T1w | 20 | 20 | – | 50-85 |

## C.2. QuickNAT

QuickNAT (Guha Roy et al., 2019) is a 2D fully convolutional framework that performs segmentation on individual slices rather than full 3D volumes. The method trains three independent F-CNNs on single coronal, axial, and sagittal slices, and fuses their outputs through a view-aggregation module to obtain the final 3D prediction. Each F-CNN adopts an encoder–decoder architecture with skip connections, unpooling layers, and dense connections to improve gradient flow and feature reuse. The network is optimized using a combination of multi-class Dice loss and weighted logistic loss to address class imbalance and enhance boundary delineation.

In its original formulation, QuickNAT is pre-trained using auxiliary labels generated by FreeSurfer and then fine-tuned on expert manual segmentations. This strategy leverages large-scale automated annotations while adapting to higher-quality ground truth. Because QuickNAT operates on single 2D slices without explicit 3D contextual modeling, its predictions may vary across views, particularly for small or irregularly shaped subcortical structures.

For our experiments, we fine-tune QuickNAT on our 7-class subcortical label set to serve as a baseline under consistent training and evaluation conditions.

## C.3. nnU-Net

nnU-Net(Isensee et al., 2021) is a self-configuring segmentation framework that automatically adapts its network architecture, data preprocessing strategies, and training pipelines to the characteristics of a given dataset. Following the original design, we relied entirely on nnU-Net's built-in mechanisms, including its automated determination of patch size, batch size, normalization scheme, deep supervision, and data augmentation policies.

For our experiments, we use the default 3D full-resolution configuration. During training, the model optimized the standard combination of Dice loss and cross-entropy loss under the framework's predefined schedule, including the default learning rate, optimizer settings, and training epochs. After training, inference was performed using nnU-Net's standard test-time augmentation and sliding-window strategy.

## C.4. MAPSeg

MAPSeg (Zhang et al., 2024) is an unsupervised domain adaptation (UDA) framework designed for volumetric medical image segmentation. It integrates 3D masked autoencoding (MAE) with a masked pseudo-labeling (MPL) strategy and a global–local consistency (GLC) objective to improve robustness across heterogeneous imaging domains. The

framework is self-supervised during pretraining through 3D MAE reconstruction, and subsequently refines pseudo-labels using MPL to adapt the model to new domains without requiring manual annotations. GLC further stabilizes training by enforcing consistency between global volumetric context and local structural details.

MAPSeg was originally proposed for centralized, federated, and test-time UDA settings, allowing models trained on one domain to generalize to unseen scanners or cohorts. Because MAPSeg operates directly on 3D volumes with domain-adaptive pseudo-labeling, it can handle cross-domain variations more effectively than purely supervised baselines.

For our experiments, we use MAPSeg's domain-adapted 3D backbone and fine-tune it on our 7-class subcortical label set to serve as a UDA-based baseline under consistent training conditions.

### C.5. SAT

SAT(Zhao et al., 2025) is a large-vocabulary medical image segmentation framework integrating an image encoder with a text-aware feature modulation module. The central design of SAT involves a text–image feature alignment mechanism and a hierarchical decoder with Mixture-of-Experts layers. The encoding process of SAT involves two encoders: a vision encoder responsible for extracting multi-scale 3D visual representations from the input medical volume, and a text encoder mapping natural language descriptions of target structures into embedding vectors. During the decoding stage, the segmentation decoder dynamically fuses these two modalities to produce structure-specific predictions.

In our experiments, we employed the SAT-nano variant built upon the nnU-Net vision backbone as recommended in the original paper. We fine-tuned the SAT-nano on our dataset using only the image domain and segmentation supervision corresponding to our task. No additional large-scale pretraining or external datasets were used.

## Appendix D. Experiment Settings

### D.1. Visual-Backbone

**MAE Pretraining.** For MAE pretraining, we follow the training configurations listed in Table 8. Each mini-batch contains a randomly sampled local patch $x$ and a downsampled global scan $X$. The masking patch size specified in Table 8 is applied only to $x$; for $X$, the masking patch is always set to half the size due to its larger field of view. During the MAE stage, we apply random 3D affine transformations with isotropic scaling between 75–150% and rotation sampled from $[-40°, 40°]$.

**MPL-GLC.** For centralized UDA brain MRI segmentation, the detailed training configurations are provided in Table 9. Each mini-batch contains four patches: a local–global pair $(x, X)$ from the source domain and another pair from the target domain (each of size $96^3$). During warm-up epochs, the model is trained exclusively on the source domain. Model selection is based on the validation *Score*, with a patience of 50 epochs.

*Target-domain augmentation.* We apply a random 3D affine transformation with isotropic scaling of 70–130% and rotation sampled from $[-30°, 30°]$.

Table 8: MAE Pretraining Configurations

| config | value |
| --- | --- |
| patch size | $96 \times 96 \times 96$ |
| local masking patch | $8 \times 8 \times 8$ |
| global masking patch | $4 \times 4 \times 4$ |
| masking ratio | 70% |
| optimizer | AdamW |
| learning rate | $2 \times 10^{-4}$ |
| weight decay | 0.05 |
| momentum | $\beta_1$=0.9, $\beta_2$=0.95 |
| lr scheduler | cosine annealing |
| epochs | 300 |
| batch size | 4 |
| iters/epoch | 500 |
| aug. prob. | 0.35 |
| augmentation | random affine |

Table 9: Fine-tuning Configurations

| config | value |
| --- | --- |
| patch size | $96 \times 96 \times 96$ |
| local masking patch | $8 \times 8 \times 8$ |
| global masking patch | $4 \times 4 \times 4$ |
| masking ratio | 70% |
| optimizer | AdamW |
| learning rate | $1 \times 10^{-4}$ |
| weight decay | 0.01 |
| momentum | $\beta_1$=0.9, $\beta_2$=0.999 |
| lr scheduler | cosine WR |
| total epochs | 100 |
| warmup epochs | first 10 |
| early stop | 50 |
| batch size | 1 |
| iters/epoch | 100 |
| augmentation | random affine |
| source aug. | random bias field |
| target aug. | random gamma |
| source prediction weight | $\beta$=0.5 |
| EMA update weight | $\alpha$=0.999/0.9999 |
| auxiliary global loss weight | $\gamma$=0.05 |
| cosine similarity weight | $\delta$=0.05 |

*Source-domain augmentation.* A stronger augmentation pipeline is used for the source domain, including random affine (70–140% scaling, $[-30°, 30°]$ rotation), random bias field, and random gamma transformation ($\gamma \in [e^{-0.4}, e^{0.4}]$).

*Teacher–student update.* The teacher model $f_\theta$ is updated using an exponential moving average (EMA) of the student parameters $f_\phi$:

$$\theta_{t+1} \leftarrow \alpha\theta_t + (1 - \alpha)\phi_t, \tag{A15}$$

where $t$ denotes the training iteration and $\alpha$ is the EMA decay rate.

*EMA scheduling.* For models initialized from large-scale MAE pretraining, we set $\alpha = 0.999$ for the first 1,000 steps and 0.9999 thereafter. For models pretrained only on small-scale datasets (tens of scans), we use $\alpha = 0.99$ for the first 1,000 steps, 0.999 for the next 2,000 steps, and 0.9999 for the remaining iterations.

The teacher network is initialized using the student parameters after a warm-up stage (e.g., 1,000 iterations) trained solely on the source domain.

### D.2. Language-Guided Prompt Encoding

The detailed training configurations of NeuroLangSeg are provided in Table 10.

## Appendix E. Results

Figure 8 shows qualitative segmentation results across methods for cross-site. Table 12 reports anatomical–linguistic evaluator scores across seven subcortical structures for in-

Table 10: NeuroLangSeg Configurations

| config | value |
| --- | --- |
| patch size | $96 \times 96 \times 96$ |
| embedding dimension | 512 |
| text max query | 32 |
| optimizer | AdamW |
| learning rate | $1 \times 10^{-4}$ |
| weight decay | 0.01 |
| momentum | $\beta_1=0.9, \ \beta_2=0.999$ |
| lr scheduler | cosine annealing |
| epochs | 500 |
| batch size | 2 |
| iters/epoch | 500 |
| shape loss weight | 0.5 |
| location loss weight | 0.5 |

Table 11: Discriminator Pretraining Configurations

| config | value |
| --- | --- |
| **shape encoder** | |
| shape embedding dimension | 128 |
| shape encoder crop size | $96 \times 96 \times 96$ |
| optimizer | Adam |
| loss | Dice+MSE |
| learning rate | $1 \times 10^{-4}$ |
| weight decay | $1 \times 10^{-5}$ |
| lr scheduler | cosine annealing |
| epochs | 50 |
| batch size | 8 |
| augmentation | gaussian noise |
| noise perturbations | 100 |
| **location encoder** | |
| location encoder batch size | 32 |
| relation pairs | 37 |
| location embedding dimension | 64 |
| optimizer | Adam |
| loss | MSE |
| learning rate | $1 \times 10^{-3}$ |
| weight decay | $1 \times 10^{-4}$ |
| lr scheduler | cosine annealing |
| epochs | 50 |
| batch size | 32 |
| augmentation | gaussian noise |
| noise perturbations | 100 |

site and cross-site CN subjects and clinical cohorts. Figure 9 shows anatomical–linguistic evaluator score distributions of seven subcortical structures across methods.

Table 12: Anatomical–Linguistic Evaluators' scores across seven subcortical structures for in-site, cross-site, and clinical generalization.*, $p < 0.05$;**, $p < 0.01$; ***, $p < 0.001$; n.s., not significant, using ANOVA with Bonferroni correction for multiple comparisons.

| Score | Methods | Seven subcortical structures | | | | | | |
|---|---|---|---|---|---|---|---|---|
| | | HIPP | AMG | CD | PT | PD | TM | AB |
| | **In-site and Cross-site (CN)** | | | | | | | |
| | FastSurfer | 88.2±2.3 | 74.8±2.4 | 73.3±1.8 | 75.9±2.1 | 74.2±1.0 | 86.0±3.7 | 74.1±6.4 |
| | QuickNAT | **94.6 ± 1.6***** | **79.9 ± 3.2***** | **71.2 ± 1.8***** | 77.4±1.7 | 74.0±1.1 | 84.9±2.5 | **70.1 ± 8.4*** |
| | nnU-Net | 89.7±1.9 | 76.5±1.2 | 74.4±1.8 | 77.6±1.6 | 73.6±0.9 | 87.3±3.2 | 76.6±5.4 |
| Shape | MAPSeg | 89.3±2.1 | 76.7±1.1 | 73.4±1.7 | 75.5±2.3 | 74.0±0.9 | 86.3±3.1 | 77.7±5.3 |
| | SAT | 90.0±2.0 | 76.0±2.0 | 73.9±1.4 | 76.5±1.6 | 73.7±0.9 | 87.6±2.7 | 78.9±5.7 |
| | NeuroLangSeg | 90.0±2.3 | 75.1±2.7 | 74.1±1.9 | 76.8±1.8 | 74.3±1.0 | 86.9±2.9 | 77.2±6.7 |
| | **Clinical generalization** | | | | | | | |
| | (ADNI) AD-NeuroLangSeg | **81.2 ± 5.7***** | **63.4 ± 3.4***** | **70.9 ± 1.6***** | **75.7 ± 0.8**** | **75.4 ± 0.6***** | **66.7 ± 3.4***** | **65.5 ± 1.6***** |
| | (BraTS) Tumor-NeuroLangSeg | **85.7 ± 3.7***** | **64.6 ± 3.0***** | **70.2 ± 2.3***** | 77.6±0.6 | 74.2±0.9 | **64.7 ± 5.6***** | **62.8 ± 1.5***** |
| | **In-site and Cross-site (CN)** | | | | | | | |
| | FastSurfer | 85.5±5.4 | 88.6±3.4 | 86.9±3.2 | 87.1±3.9 | **88.6 ± 2.6***** | 87.1±4.2 | 87.2±3.2 |
| | QuickNAT | **82.7 ± 3.7***** | **79.1 ± 9.5***** | **81.8 ± 4.6***** | **79.6 ± 3.3***** | **77.0 ± 10.2***** | **77.2 ± 8.6***** | 82.4±4.2 |
| | nnU-Net | 84.8±5.0 | 88.0±4.3 | 85.4±4.0 | 86.5±4.7 | 87.8±4.3 | **85.9 ± 4.7**** | 86.0±4.6 |
| Location | MAPSeg | 86.3±3.7 | 89.0±2.9 | 87.3±3.0 | 88.1±2.7 | **88.9 ± 2.6***** | 87.8±3.3 | 87.8±3.0 |
| | SAT | 85.8±3.7 | 88.2±3.2 | 85.8±3.4 | 86.3±3.5 | 87.2±3.5 | 87.4±3.2 | 86.4±3.9 |
| | NeuroLangSeg | 86.9±3.1 | 89.7±2.5 | 87.9±2.7 | 86.2±2.4 | 85.7±2.3 | 89.8±2.8 | 85.8±4.0 |
| | **Clinical generalization** | | | | | | | |
| | (ADNI) AD-NeuroLangSeg | **69.3 ± 4.9***** | **80.2 ± 5.1***** | **70.6 ± 3.4***** | **74.7 ± 3.9***** | **74.3 ± 3.4***** | **74.0 ± 4.2***** | **73.0 ± 5.3***** |
| | (BraTS) Tumor-NeuroLangSeg | **70.0 ± 5.7***** | **73.9 ± 6.2***** | **67.6 ± 9.0***** | **66.1 ± 6.9***** | **67.0 ± 6.7***** | **70.5 ± 11.7***** | **62.7 ± 10.9***** |
| | **In-site and Cross-site (CN)** | | | | | | | |
| | FastSurfer | **1.46 ± 0.96**** | **2.25 ± 1.05***** | **1.40 ± 1.01**** | 1.02±0.77 | 1.79±1.59 | 0.75±0.66 | 1.06±0.75 |
| | QuickNAT | **1.55 ± 1.10**** | **3.56 ± 1.34***** | **1.69 ± 1.13***** | 0.84±0.57 | **2.54 ± 0.94***** | **1.53 ± 0.79***** | **3.50 ± 1.02***** |
| | nnU-Net | 1.10±0.70 | **1.93 ± 0.89*** | 0.82±0.60 | 0.75±0.54 | 1.29±0.76 | 0.67±0.52 | 0.79±0.61 |
| Volume | MAPSeg | 1.01±0.68 | 1.40±0.68 | 1.13±0.80 | 0.94±0.65 | 1.28±0.82 | 0.70±0.55 | 0.71±0.49 |
| | SAT | 0.98±0.80 | **1.92 ± 0.90*** | 1.07±0.69 | 0.68±0.54 | 1.36±1.03 | 0.74±0.60 | 0.86±0.60 |
| | NeuroLangSeg | 0.91±0.67 | 1.44±0.77 | 0.84±0.55 | 0.72±0.50 | 1.13±0.74 | 0.78±0.54 | 0.77±0.59 |
| | **Clinical generalization** | | | | | | | |
| | (ADNI) AD-NeuroLangSeg | **2.30 ± 1.27***** | **2.88 ± 1.23***** | **1.62 ± 1.02***** | **1.41 ± 0.84***** | **2.80 ± 0.87***** | 0.81±0.54 | 0.93±0.78 |
| | (BraTS) Tumor-NeuroLangSeg | **2.12 ± 1.29***** | **3.64 ± 1.97***** | 1.20±1.41 | **1.59 ± 1.53***** | **3.58 ± 1.46***** | **1.70 ± 1.87***** | **1.83 ± 1.71***** |

HIPP:Hippocampus, AMG:Amygdala, TM:Thalamus, CD:Caudate, PT:Putamen, PD:Pallidum, AB:Accumbens

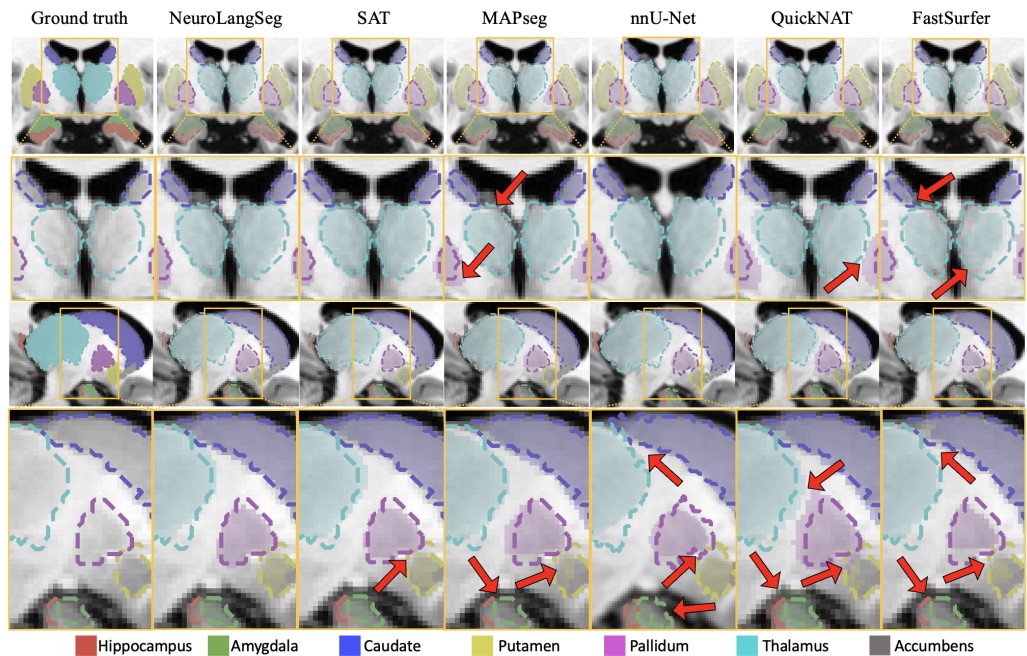

Figure 8: Qualitative segmentation performance for cross-site. Coronal and sagittal views are shown, together with corresponding zoomed-in regions of interest. Major segmentation errors are highlighted with red arrows. Ground-truth boundaries are indicated by dotted lines, while segmentations from different methods are shown as transparent overlays.

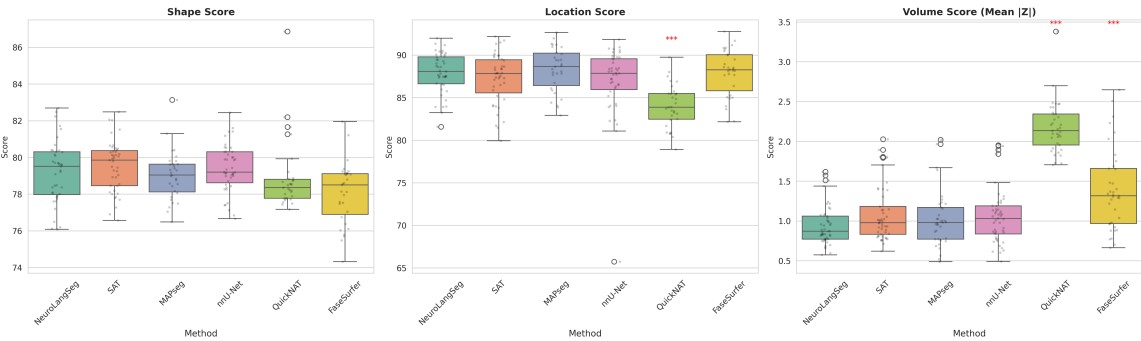

Figure 9: Shape, location, and Z-score distributions of seven subcortical structures compared to baselines. *** indicate statistically significant differences with $p < 0.001$ under Bonferroni correction.

