# OpenReview forum: "NeuroLangSeg: Language-Guided Subcortical Segmentation with Pseudo-Supervision and Anatomical–Linguistic Validation"
_MIDL.io/2026/Conference — MIDL 2026 Poster_

### Official Review · Reviewer_kyxN · 2026-01-08

**Confidence:** 3
**Preliminary Rating:** 3
**Final Rating:** 4

**Summary:**

This paper introduce NeuroLangSeg, a method for brain MRI segmentation which incorperated contextual anatomical reasoning and enforces a consistent anatomical protocal for subcortical segmentation.  The framework combines a pretrained image encoder, protocol-aligned anatomical prompts, and a masked pseudo-labeling strategy to enable data-efficient and interpretable learning under limited annotations.

**Strengths:**

1. The paper tackles the critical issue of inconsistent anatomical protocols across datasets, which is a major obstacle for reliable deployment of segmentation models in clinical applications.
2. The authors evaluate the method across in-site, cross-site, and disease-cohort settings, providing a broad assessment of performance under realistic conditions.
3. The proposed Anatomical–Linguistic Discriminator is a novel contribution that explicitly incorporates anatomical awareness into the training process.

**Weaknesses:**

1. The description of the topological discriminator lacks sufficient detail regarding how the anatomical protocol is obtained and formalized.
2. The protocol-derived topological constraints appear to encode canonical anatomical relationships but do not explicitly account for inter-subject variability due to age, development, or pathology
3. The same anatomical protocol is used to guide model training and to evaluate predictions using anatomical–linguistic metrics. This raises concerns about evaluator circularity

**Detailed Comments:**

1.The paper would benefit from a clearer and more explicit description of how the anatomical protocol is constructed and operationalized.

**Justification Of Final Rating:**

The author's reply addresed my questions, I would like to raise the score to 4. The paper proposes a novel and well-motivated approach that addresses an important challenge in medical image segmentation, integrating language-guided learning with anatomical protocol enforcement.

**Justification Of The Preliminary Rating:**

The paper proposes a novel and well-motivated approach that addresses an important challenge in medical image segmentation, the inconsistency of anatomical protocols across datasets, which is highly relevant for clinical applications. However several aspects of the approach needs to be clarified.

**Questions To Address In The Rebuttal:**

1. Can the authors clarify how the anatomical protocol text is obtained and formalized?
2. How does the proposed topological discriminator account for natural anatomical variability across subjects, ages, or pathological conditions?

---

> ### Author Response · Authors · 2026-01-24
> **Response to reviewer comment**
>
> **Response to Weakness 1**: We thank the reviewer for pointing out the need for greater clarity. We have revised Section 2.3.2 in the manuscript to provide a detailed description of how the protocol is obtained from Neuromorphometrics, Inc. and formalized. Specifically, we explain the process of using an LLM to extract the anatomical pairs and converting text description into continuous feature vectors. We used the following prompt to extract anatomical rules into a JSON format: "Please extract the morphological features, relevant reference regions for manual annotation, and positional relationship descriptions... and convert them into a structured JSON description." The LLM output identified 37 key anatomical pairs (15 left, 15 right, 7 cross-hemisphere) and defined their relational types in a structured JSON format.
>
> For example, from the sentence “the hippocampus is posterior and inferior to the amygdala” in protocol, the LLM outputs structured JSON: {hippocampus-amygdala: {relative_position: [-1, -1, 0], adjacency_ratio: 1,adjacency_vector}: [1, 1, 0]}. The detailed procedure for extracting the relational features is provided in Appendix A7.
>
> **Response to Weakness 2**: Thank you for the comment. We **normalize the topological features to reduce the effect of age-related brain size and developmental differences**, so the constraints do not depend on absolute brain scale. This allows the same topological rules to be applied across subjects of different ages.
>
> The discriminator is **trained on healthy subjects** to learn a reference of normal anatomy. When applied to AD or tumor cases, lower discriminator scores indicate abnormal anatomical organization, which is why we explicitly evaluate the method on these populations.
>
> **Response to Weakness 3**: We thank the reviewer for raising this concern. We clarify that the anatomical–linguistic **discriminators are not optimized jointly** with the segmentation model. They are **trained once** using manual labels and a fixed anatomical protocol, and are **frozen** during segmentation training and evaluation. The segmentation network does not receive gradients from the discriminators. This clarification has been added in the revised manuscript (Section 2.4).
>
> Importantly, the anatomical–linguistic discriminators are not designed to measure voxel-wise segmentation accuracy. Instead, they assess whether a predicted segmentation conforms to normal anatomical rules, analogous to how BrainChart evaluates whether an individual is anatomically typical or atypical, rather than how closely a segmentation matches a voxel-level reference.
>
> The scores of CN subjects using NeuroLangSeg is better. This improvement is not an artifact of training, as the discriminators are fixed and protocol-driven. As discussed in Section 4.3.2, most methods show no significant differences when evaluated on the same CN cohort; only QuickNAT and FastSurfer differ significantly (p < 0.001), consistent with their lower DSC/NSD performance. In contrast, when applying NeuroLangSeg across CN, AD, and tumor groups, all three evaluators show significant differences (p < 0.001), indicating stability in healthy controls and sensitivity to disease-related anatomical changes.

---

### Official Review · Reviewer_Dssy · 2026-01-09

**Confidence:** 4
**Preliminary Rating:** 2
**Final Rating:** 3

**Summary:**

The proposed method, NeuroLangSeg, is a language-guided framework to achieve subcortical anatomical segmentation. The framework builds on the existing framework of MAPSeg, and enforces a consistent anatomical labeling protocol. The main contributions include a prompt-conditioned segmentation and a unified visual backbone (using MAE pretraining, pseudo-label refinement, global-local stabilization). The training data set covers 1-100yrs age range and includes multimodal (T1/T2) input images. The experiments investigate multiple aspects of the new framework. The outcomes are promising.

**Strengths:**

Evaluation is computed via multiple metrics:  via morphological, volumetric metrics and anatomical consistency measures.

The performance metrics are compared to visual-only and vison-language models. In-site, cross-site comparison and evaluation on clinical data is also used.

The proposed method has performed with a +0.3% DSC and +1.0% NSD gain over the strongest baseline. The cross-site generalization is results strong.

Code has been made available

**Weaknesses:**

The segmentation accuracy performance gain does not seem very significant. (No statistical significance over Dice overlap values, for example).

It is not clear how all baseline methods were selected. For example, why wasn't the popular SynthSeg model used? (The inclusion of nnUNet and MAPSeg are covered.)

Ablation study is missing in which loss term managed to add the most to the current method.

The manual segmentation protocol details are missing from the ADNI data.

**Detailed Comments:**

Overall decent writing quality. This could be improved by a full round of spell-check, proper acronym definitions, better citation description

Some citations are not properly used, or the surrounding text does not fully correspond to the statements made around them. For example,
* I am not sure why Tae 2008 is cited with Freesurfer-based segmentations.
* re Lerch 2017: This paper discusses: "MRI provides an indirect measurement of the biological signals we aim to investigate".  This is different from the statement that Freesurfer diverges form expert manual labels.
Clarifications around these would be welcome.

One main motivation of the authors to develop this new method is the lack of unified anatomical definition of structures to be segmented. I feel this issue has not been addressed. The method is presented with a single set of segmentation guidelines, but that does not help resolve discrepancies / disagreements between anatomical definitions. The method refers to Neuromorphometics's segmentation rules -- this is still just one of the definitions.

Multiple brainchart citations are provided (Rutherford and Bethlehem): Are there any conflicting numbers in these -- if yes, how are the potential discrepancies mitigated? Are there any limitations that you could discuss related to these? Age range / race info/ ...?

More details would be welcome regarding who segmented the N = 119 hand-picked ADNI images. Also what protocol was used and how many labels defined? Was it a full brain segmentation or only a 7-label segmentation?

Accuracy is not shown in 4.3.2; instead ANOVA analysis comparing CN and pathological cases

Additionally:
=========
make sure to capitalize "Dice" loss
FSS is never defined
2.3.1 word is missing after "Different subcortical"
is radio == ratio?
In 2.3.2: How is K defined?
Define acronyms at their first appearance
Is there a difference between Dice loss and DSC that only gets defined in 3.1 (while Dice loss is referenced in earlier sections as well)
Table 1: how were the 7 subcortical structures selected?

**Justification Of Final Rating:**

I would like to thank the authors for their detailed rebuttal and providing additional information to support their submission.  I move my score to borderline due to all the clarifications that have been made to the manuscript.

**Justification Of The Preliminary Rating:**

The proposed method addresses important segmentation challenges in medical imaging. The formulation of the pipeline and the description of various design choices regarding the setting up of the experiments and the characterization of the outcomes however lacks some rigor, which should be improved upon.

**Questions To Address In The Rebuttal:**

How does this framework mitigate expert differences when defining anatomical boundaries?

Future wok calls for an extension to infant and pediatric cases. However the current age range encompasses an age range from 1yr. What will be the age range for infant and pediatric subjects?

Can you clarify what "anatomically verifiable predictions" mean? Why wouldn't competing methods deliver the same type of predictions?

Clarification: Related to the shape consistency term -- Is there no variance information encoded here?

In Introduction. Can you clarify statement regarding methods such as SynthSeg and VINNA learning labels directly from MRI images? Both of these methods use labels as inputs for their training.

Fig 2: Pallidum seems a bit over-segmented is in the proposed method. Would you comment?

---

> ### Author Response · Authors · 2026-01-24
> **Response to reviewer comment**
>
> **Weakness 1:** We agree with Dice improvement is not significant. This is expected. Methods such as FreeSurfer already achieve high Dice scores (about 0.85–0.88) (Woo Suk Tae, 2008). So there is little room for large gains. Our main contribution is not higher Dice, but a unified anatomical protocol and evaluation framework for settings where labels are missing or inconsistent. Dice requires manual annotations—often unavailable in real clinical practice—and cannot assess anatomical plausibility. Our anatomical–linguistic discriminator instead evaluates whether segmentations follow normal anatomical rules, enabling cross-dataset and unlabeled evaluation where Dice is not applicable.
>
> **Weakness 2:** nnU-Net is a strong standard 3D segmentation baseline. MAPSeg is our prior work and serves as the backbone-based baseline. QuickNAT represents 2D slice-based segmentation, while FastSurfer is a widely used deep-learning alternative to FreeSurfer; we use its latest VINN architecture with built-in resolution augmentation. SynthSeg is a strong method but is trained on simulated whole-brain T1/T2 data and designed for full-brain segmentation, making it not directly applicable to our subcortical-focused setting. **More detailed specifications are provided in the Section 4 experimental setup**.
>
> **Weakness 3:** We have added additional ablation experiments in Table 2 in Section 4.3.
>
> **Weakness 4:** The manual segmentation labels are provided by Neuromorphometrics, Inc, developed under Dr. Andrew Worth. They also provide label for **FreeSurfer** (https://www.neuromorphometrics.com/?page_id=219), indicating that this protocol is trusted by existing clinical segmentation pipelines. The **ADNI** manual labels are part of the labeled dataset: https://neuromorphometrics.com/2016-03/LabeledScans.html.
>
> For the ADNI, CANDI, Colin, and OASIS datasets, performed whole-brain segmentation. For the BCP dataset, only seven subcortical structures were manually labeled (NIMH Data Archive (Study #1745)).
>
>
> **Citation Tae 2008:** Prior work has demonstrated that FreeSurfer achieves Dice scores in the range of 0.85–0.88 for subcortical structures, as reported by Tae et al. (2008). It is also cited by SynthSeg (page 6), where FreeSurfer is used as a reference.
>
> **Citation Lerch 2017:**  Lerch et al. (2017) provides a conceptual framework on systematic bias in automated MRI analysis, as also noted in QuickNAT (Roy et al., 2018). We clarified this in the manuscript and added empirical studies (Morey et al., 2009; Schoemaker et al., 2016) showing FreeSurfer’s divergence from manual labels, along with references (Geuze et al., 2005; Yushkevich et al., 2015) documenting the lack of a unified clinical delineation protocol.
>
> **Response to protocol:** Please see answer 4 and 1. Our goal is build a model, discriminator, and evaluation framework that operate under a fixed, unified protocol (create by Neuromorphometrics). If users prefer a different anatomical protocol, the same discriminator and evaluation design can be applied to any new unified labeling standard by retraining the evaluator on that protocol.
>
> Segmentation accuracy (Dice) for healthy controls (CN) is reported in Table 1. **Dice scores for AD and tumor cases are not available** because manual labels do not exist for these cohorts.
>
> **BrainChart:** Both models are derived from FreeSurfer. Bethlehem et al. is based 123,984 subjects, but does not report for some subcortical structures below early childhood (e.g., thalamus before 6 years). Rutherford et al. uses 58,836 subjects and provides complete normative trajectories for all seven subcortical structures across a consistent age range (0–100 years). For this reason, we adopt Rutherford et al. as reference.
>
> **Additionally**: Dice capitalization: Corrected. FSS : Defined in Section 2.3.1. Section 2.3.1 wording: corrected “radio” to “ratio.”Definition of K: Explicitly defined as the number of anatomically paired structures, with further details in the Appendix (Table 4). Dice loss vs. DSC: Clarified that Dice loss = 1 - DSC at (Zhang et al., 2024). Subcortical definition: Following common neuroimaging conventions (e.g., FreeSurfer, BrainChart, Mamah et al. (2016))
>
> **Age**: We use 1–2 year data for training because this mainly requires adding more samples, and brain development is more stable than in 0–1 year. The current paper does not include infant or pediatric clinical cohorts in Table 3, although infant data are used for training. Pediatric clinical populations will be studied in future work.
>
> The shape features are learned from healthy subjects across ages and are normalized to reduce effects of rotation, scale, and orientation. This allows natural variability while making large pathological deformations stand out as deviations from the learned healthy reference.
>
> Minor 1–2 pixel over-segmentation may still occur, which is a common and well-known behavior in deep learning–based segmentation methods.

---

### Official Review · Reviewer_Ezys · 2026-01-09

**Confidence:** 4
**Preliminary Rating:** 3
**Final Rating:** 3

**Summary:**

This paper introduces NeuroLangSeg, a framework for subcortical brain MRI segmentation that integrates language-guided learning with anatomical protocol enforcement. The key contributions include: (1) a unified visual backbone combining 3D masked autoencoder pretraining with pseudo-label refinement, (2) language-guided prompts for structure-specific segmentation, and (3) an anatomical-linguistic evaluator that assesses morphological, topological, and volumetric consistency during training and inference.

**Strengths:**

1. The motivation is clear and is directly targeting at an existing problem that current segmentation methods are constrained by heterogeneous labeling protocol. The idea of creating a consistent anatomical protocol is clinically meaningful.
2. The framework design is comprehensive with integration of multiple components. The three evaluators (morphological, topological, volumetric) give an interpretable view of the segmentation results beyond traditional metrics.
3. Experimental results are quite impressive, with superior DSC/NSD performance both in-site and cross-site.

**Weaknesses:**

1. The sample size of cross-site evaluation is somewhat low with only one subject,
2. The anatomical-lingusitc discriminators are used both in training and evaluation. The model could be optimized to achieve good scores on these discriminators during training to get good performance when evaluating on the same models.
3. No ablations on the effectiveness of each component in the framework.

**Detailed Comments:**

See above.

**Justification Of Final Rating:**

I thank the authors for their clarification of their methods and the ablations. However, I believe these problems are pretty fundamental for the initial submission of a paper. I would like to keep my original score.

**Justification Of The Preliminary Rating:**

This is a solid contribution with novel ideas around protocol-consistent segmentation and anatomical-linguistic evaluation. However, the cross-site evaluation is limited, evaluation metrics are weak and ablation studies are needed.

**Questions To Address In The Rebuttal:**

See above.

---

> ### Author Response · Authors · 2026-01-24
> **Response to reviewer comment**
>
> **Response to Weakness 1:** Thank you for pointing this out, and we apologize for the confusion in the original description. We clarify the site setting as follows. In our study, **CANDI, Colin, and OASIS are single-site datasets**, whereas **ADNI manual dataset includes 30 subjects spanning 25 distinct sites**.
>
> ADNI subjects are split such that 15 subjects from 12 sites are used for training, 3 subjects from 2 sites are used for validation, and 12 subjects from 11 different sites are used for testing.  Colin and ADNI 12 testing subjects reserved exclusively for cross-site inference. Therefore, **all ADNI test subjects come from sites unseen during training**, instead of a single external dataset (Colin).
>
> **We have corrected Table 1 and Experiment 2 in Section 4.2, and provided additional clarification in Appendix B.**
>
> **Response to Weakness 2:** We thank the reviewer for raising this concern. We clarify that the anatomical–linguistic discriminators **are not optimized jointly with the segmentation model**. They are **trained once** using manual labels and a fixed anatomical protocol, and are **frozen** during segmentation training and evaluation. The segmentation network does not receive gradients from the discriminators. **This clarification has been added in the revised manuscript (Section 2.4)**.
>
> Importantly, the anatomical–linguistic discriminators are not designed to measure voxel-wise segmentation accuracy. Instead, they assess whether a **predicted segmentation conforms to normal anatomical rules**, **which is a core novelty of our method**. Like how BrainChart evaluates whether an individual is anatomically typical or atypical, rather than how closely a segmentation matches a manual label.
>
> In Table 3, *** denotes statistically significant differences (p < 0.001). These differences arise from two distinct factors:
> (1) low segmentation quality in certain baseline methods (e.g., QuickNAT), and
> (2) anatomical abnormalities present in clinical cohorts (AD and tumor).
>
> The scores of CN subjects using NeuroLangSeg is better. This improvement is not an artifact of training, as the discriminators are fixed and protocol-driven. As discussed in Section 4.3.2, most methods show no significant differences when evaluated on the same CN cohort; only QuickNAT and FastSurfer differ significantly (p < 0.001), consistent with their lower DSC/NSD performance. In contrast, when applying NeuroLangSeg across CN, AD, and tumor groups, all three evaluators show significant differences (p < 0.001), indicating stability in healthy controls and sensitivity to disease-related anatomical changes.
>
> **Response to Weakness 3:** Thank you for pointing out the lack of ablation studies in the original submission. Following this suggestion, we have conducted additional ablation experiments on the in/cross-site dataset to systematically analyze the contribution of each discriminator in our framework. **The results are reported in Table 2 in Section 4.3.**

---

### Author Rebuttal · Authors · 2026-01-24

**Rebuttal:**

We thank all reviewers for their careful reading and constructive feedback. We have revised the manuscript to address each reviewer’s comments as follows.

**Reviewer Ezys.**  We corrected Table 1 to clearly distinguish in-site and cross-site evaluation and moved ADNI from the in-site to the cross-site setting. This directly addresses the concern that cross-site evaluation involved too few subjects and corrects the previously reported performance scores. The revised table and text now accurately reflect the experimental design and results.

**All Reviewers.** We added Table 2 to present a dedicated ablation study, clarifying the contribution of each component in the proposed framework. Additional details on model configuration and experimental settings have also been added for clarity.

**Reviewer Dssy.**  We revised the qualitative segmentation comparisons to improve interpretability: ground-truth boundaries are shown with dotted lines, and segmentations from different methods are displayed as transparent overlays. We also clarified typical over-segmentation behavior in deep learning methods to address the reviewer’s concern.

During this review, we identified one repeated ADNI subject; after removal, the total number of manual labels was corrected. Finally, the total number of manual annotations has been corrected from 119 to 118.

In addition, we incorporated further clarifications throughout the revised manuscript regarding model definitions, evaluation protocols, and experimental settings, addressing all remaining reviewer questions. **Please refer to the revised manuscript for full details.**

**Supporting Material:**

/attachment/d846b577e7acd6aa7c0ef316db4248a5b4340c5a.pdf

---

### Meta-Review · Area_Chair_F7hQ · 2026-02-05

**Recommendation:** Accept (Poster)
**Confidence:** 4

**Metareview:**

Interesting paper which tries to embed textual context into segmentation. I am particularly interested in the anatomical-linguistic part with the evaluations.

To me, the authors addressed the reviewers concerned adequately. I think the ablation experiments is particularly important. The average score is > 3. Thus, I choose acceptance.

---

### Decision · Program_Chairs · 2026-02-13

Accept (Poster)